# Ultra-high precision nano additive manufacturing of metal oxide semiconductors via multi-photon lithography

Chun Cao [1,5], Xianmeng Xia[2,5], Xiaoming Shen[3,5], Xiaobing Wang[2], Zhenyao Yang[2], Qiulan Liu[4] ✉, Chenliang Ding[4], Dazhao Zhu[4], Cuifang Kuang [3,4] ✉ & Xu Liu [3,4]

As a basic component of the versatile semiconductor devices, metal oxides play a critical role in modern electronic information industry. However, ultra-high precision nanopatterning of metal oxides often involves multi-step lithography and transfer process, which is time-consuming and costly. Here, we report a strategy, using metal-organic compounds as solid precursor photoresist for multi-photon lithography and post-sintering, to realize ultra-high precision additive manufacturing of metal oxides. As a result, we gain metal oxides including ZnO, CuO and $ZrO_2$ with a critical dimension of 35 nm, which sets a benchmark for additive manufacturing of metal oxides. Besides, atomic doping can be easily accomplished by including the target element in precursor photoresist, and heterogeneous structures can also be created by multiple multi-photon lithography, allowing this strategy to accommodate the requirements of various semiconductor devices. For instance, we fabricate an ZnO photodetector by the proposed strategy.

Semiconductor chips following Moore's Law require that lithography technologies evolve with the times. Although state-of-the-art deep ultraviolet (DUV) and extreme ultraviolet (EUV) lithography enable high-throughput fabrication of nanostructures, expensive masks, and repetitive exposure transfer are inevitable, which is inaccessible to preliminary scientific exploration[1]. As an alternative strategy, mask-less multi-photon lithography (MPL) allows the fabrication of arbitrary 2D/3D nano-architectures and is cheaper, more available, and atomically economical than traditional strategies[2–5], especially in high-precision, small-batch manufacturing[6–9]. However, there are still many challenges to overcome for MPL, and two of the key issues are the undesirable critical dimension (CD) and the scarcity of photoresist libraries[10]. To date, most photoresists used for MPL consist of liquid organic polymers that typically have a CD limit of 80–100 nm and struggle to withstand harsh pattern transfer processes, which can be fatal for chips and device applications[11,12]. In response, some scientific pioneers have changed their minds by developing precursor photoresists and directly realizing nano-additive manufacturing of target semiconductors and other functional materials based on MPL[13–17]. As a result, complex transfer processes can be completely eliminated. Among them, metal−oxide semiconductors (MOS) have received considerable attention since they are widely used in the next generation of microelectronic devices, including transistors[18], memristors[19], optoelectronics[20], and sensors[21].

Precursor photoresists are critical for additive manufacturing of MOS via lithography techniques. Typically, precursor photoresists for

[1]School of Mechanical Engineering, Hangzhou Dianzi University, Hangzhou 310018, China. [2]Research Center for Astronomical Computing, Zhejiang Lab, Hangzhou 311121, China. [3]ZJU-Hangzhou Global Scientific and Technological Innovation Center, Hangzhou 311200, China. [4]State Key Laboratory of Extreme Photonics and Instrumentation, College of Optical Science and Engineering, Zhejiang University, Hangzhou 310027, China. [5]These authors contributed equally: Chun Cao, Xianmeng Xia, Xiaoming Shen. ✉e-mail: 21430066@zju.edu.cn; cfkuang@zju.edu.cn

MOS can be divided into three types. The first type is the metal oxide precursor, in which the metal oxide nanoparticles are simply mixed with active resins, and the target MOS can be achieved by MPL and sintering. At this point, the inhomogeneous size of the nanoparticles and their aggregation in the active resin will lead to defects in the MOS pattern[15,22]. Meanwhile, nanoparticles will cause significant light scattering and micro-explosion, giving rise to worse CD and line edge roughness. The second type is metal ion precursors. To solve the above problems, another strategy is to mix metal ions in the active resins and pyrolyze it after MPL to form MOS patterns[23,24]. However, this brings new disadvantages. One of them is the low metal ion content, as they have finite solubility in organic resins (or hydrogel resins) and are lost during the development process, resulting in high shrinkage and poor pattern quality[25,26]. More importantly, metal ions, especially transition metal ions, tend to have redox capabilities and could quench active free radicals, thus weakening or even terminating the polymerization in MPL. The third type is metal-organic compounds (MOCs). MOCs seem to bypass the above shortcomings as they can be blended with a wide range of reactive resins at the molecular level with sizable loadings. Recently, Bauer et al. reported the fabrication of high-quality fused silica glass nanostructures from polyhedral oligomeric silsesquioxane (POSS) resin via MPL[27]. Meanwhile, Xiong et al. introduced metal–organic frameworks into active resins and successfully realized the MPL of 3D MOS, including ZnO and $Co_3O_4$[28]. Besides, Malinauskas et al. employed hybrid organic-inorganic material SZ2080 in MPL to achieve Zr/Si oxides[29]. Unfortunately, the above MOS precursor photoresists are still in liquid form, which is more suitable for the construction of 3D nanostructures but difficult to realize ultra-high resolution MOS patterns via MPL. Due to the high migration velocity of free radicals hindering the possibility of CD improvement[30,31], the resulting CD is hardly less than 60 nm[32], thus failing to meet the requirements of highly integrated chips and devices. In general, the migration velocity of free radicals decreases as the shear viscosity increases[33–35]. In addition, the photoresists used in current mature photolithography technologies are all solid films, allowing for fast, high-resolution, and large-area exposures[36,37]. Therefore, solid-state precursor photoresists may be more advantageous to form high-resolution MOS patterns.

Herein, we report a universal strategy to achieve multiple MOS patterns with ultra-high precision via MPL by using active MOC-based solid precursor photoresists. Without complex synthesis steps, we chose commercially available acrylic metal complexes as resins, which can not only form excellent solid films by spin-coating but also highly polymerize by MPL to generate target patterns. To further suppress the migration of free radicals, a free radical trapping agent (quencher), bis(2,2,6,6-tetramethyl-4-piperidyl-1-oxyl) sebacate (abbreviated as BTPOS), is introduced into the solid precursor photoresists, by which the excess free radicals will be well confined in a narrower space, thus favoring the improvement of CD[38,39]. Based on this strategy, typical MOS including zinc oxide (ZnO), copper oxide (CuO), and zirconium dioxide ($ZrO_2$), are successfully obtained by MPL and subsequent pyrolysis. Remarkably, all three metal oxide patterns have CDs below 100 nm, with zinc oxide reaching 35 nm, which is comparable to the conventional strategy. Besides, the possibility of atomic doping in MOS pattern is explored by hybriding the trace element in solid precursor photoresists, Notably, this is very significant for the performance enhancement and application expansion of MOS[40,41]. Meanwhile, heterostructure MOS patterns are realized by MPL, proving our strategy can satisfy the additive manufacturing of complex micro-devices. Finally, we fabricate a ZnO photodetector by our strategy. Overall, this work sets a benchmark for ultra-high precision nano additive manufacturing of metal oxide semiconductors, and may pay the way for MPL toward nano additive manufacturing of highly integrated chips and devices.

## Results

### Patterning metal oxides using solid precursor photoresists

As shown in Fig. 1a, the solid precursor photoresists consist of acrylic metal complexes, initiator (DETC), and free radical trapping agent (BTPOS). Figure 1b shows the step-by-step schematic diagram of the additive manufacturing process of MOS via MPL to prepare the photoresist films, acrylic metal complexes, initiator, and BTPOS are dissolved in a solvent (Fig. 1c and Supplementary Fig. 1), and then spin-coated on the substrate. Although the resultant photoresist films (approx. 30 nm in thickness) inherit the color of the metal ions and DETC (Supplementary Fig. 2), they still have good transmittance to the wavelength (525 nm) of the used laser (Supplementary Figs. 3–5), which can ensure the feasibility of MPL. During the MPL process, DETC in the exposure area can generate a large number of free radicals through multi-photon absorption, which in turn triggers the polymerization of the carbon-carbon double bonds in the precursor photoresist, leading to a significant solubility difference from the unexposed photoresist, and ultimately the target pattern is obtained after development. After MPL development, and sintering, the targeted 2D micropatterns are successfully realized using Zn, Cu, and Zr-based acrylic metal complexes, respectively (Fig. 1d–f). Obviously, the fabricated Chinese characters from Zn/Cu/Zr-based precursor photoresists are highly recognizable, dense, and free of pores. Meanwhile, the sintering process does not lead to any significant deformation or distortion of the generated line arrays, regardless of the acrylic metal complex used (Fig. 1d–f), indicating that the present strategy is versatile and potentially capable of realizing high-precision MOS.

The composition of the micropatterns is confirmed by energy-dispersive X-ray spectroscopy (EDS), transmission electron microscope (TEM), and X-ray diffraction (XRD). Prior to sintering, the micropatterns generated by the Zn-based precursor photoresist mainly contain Zn, O, and C elements (Fig. 1g), while after sintering, it is difficult to recognize the C element (Fig. 1h), suggesting that the organic components are removed during the sintering process. EDS spectrum (Fig. 1i) shows the atomic percentages of the micropatterns after sintering, the respective atomic ratios of Zn and O are 31.11% and 31.13%, i.e., Zn/O = 1.00, which is consistent with the atomic ratio of the ZnO crystal. Notably, trace C (atomic ratio: 2.30%) is mainly from combustion residue and SEM sediment contamination[23], Si derives from the silicon substrate, and Pt comes from the sputtered conductive layer on the sample surface. The TEM selected area electron diffraction (SAED) patterns of pyrolyzed Zn-based precursor photoresist (Fig. 1j) exhibit the characteristic rings of ZnO crystals, and high-resolution transmission electron microscopy (HRTEM) images also provide two typical space lattices (0.287 nm (100), and 0.267 nm (002)) of ZnO crystals (Fig. 1k, l), proving that the main component after sintering should be ZnO. In addition, all the sharp diffraction peaks observed from the XRD spectra of Fig. 1m are in good agreement with the standard polycrystalline zinc oxide card (JCPDS No. 36-1451), and no other peaks (such as the graphitized carbon at 26°) are observed, indicating that no by-products or impurities are generated, and the final micropatterns are composed of ZnO. Likewise, using Cu or Zr-based precursor photoresist, the successful additive manufacturing of CuO and $ZrO_2$ micropatterns is confirmed by EDS (Supplementary Figs. 6 and 7) and XRD (Fig. 1n, o). As shown in Fig. 1n, all the sharp diffraction peaks observed are consistent with standard polycrystalline CuO card (JCPDS No. 45-0937), and no other peaks are observed, which indicate that the product is polycrystalline CuO. Unlike ZnO and CuO, the crystalline form of $ZrO_2$ depends on the sintering temperature (Fig. 1o), and at 550 °C the tetragonal phase (JCPDS No. 49-1642) is predominant. As the sintering temperature increases to 800 °C, the tetragonal phase gradually transforms into a monoclinic phase (JCPDS No. 37-1484), and finally to a completely monoclinic phase at 1000 °C[42,43]. Overall, this strategy enables the realization of MOS micropatterns, including ZnO, CuO, and $ZrO_2$.

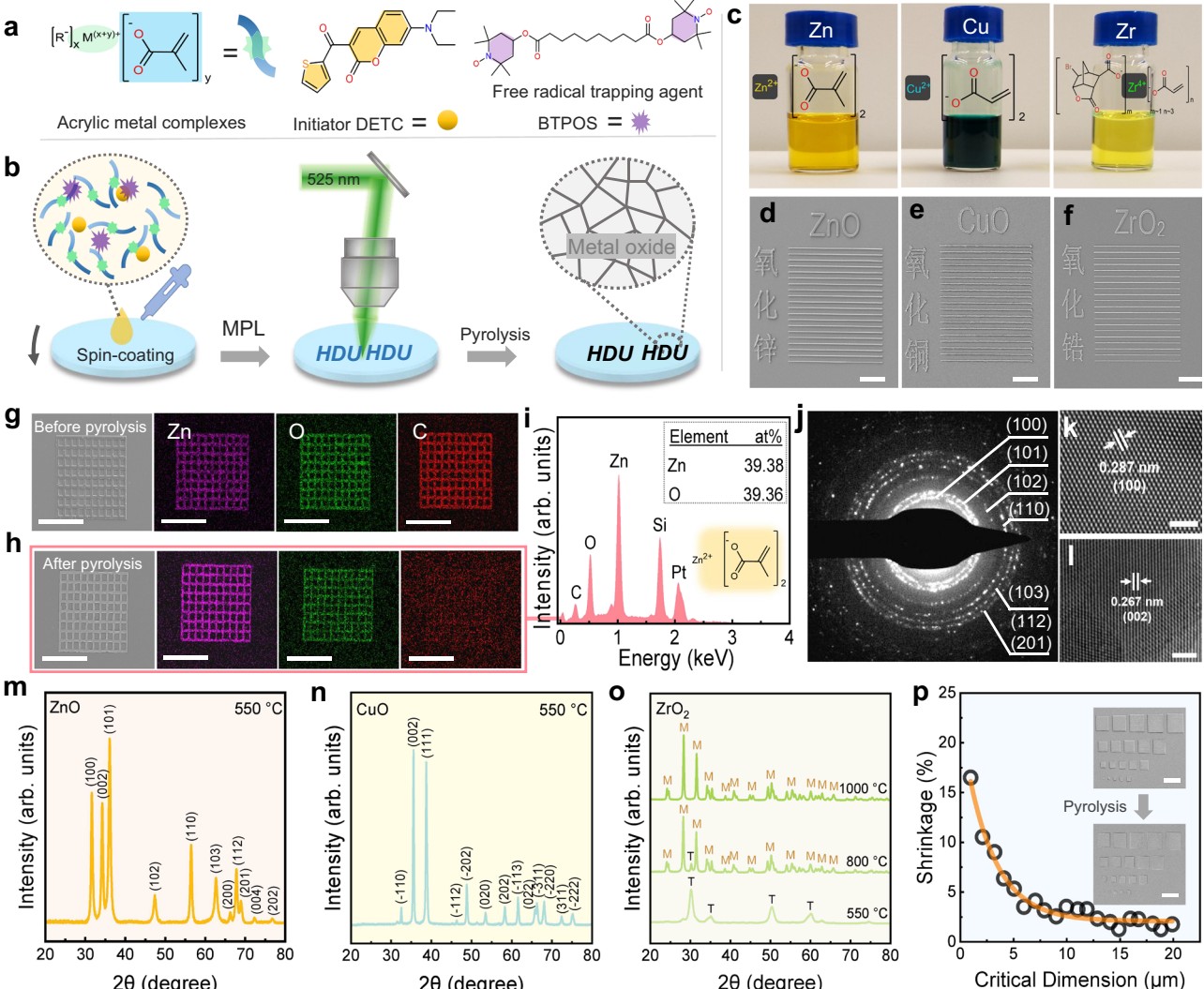

**Fig. 1 | Fabricating MOS micropatterns based on MPL. a** Chemical components of the precursor photoresists: acrylic metal complexes, initiator (DETC), and free radical trapping agent (BTPOS). **b** Step-by-step schematic diagram of the additive manufacturing process of MOS via MPL. HDU stands for Hangzhou Dianzi University. **c** The images of different acrylic metal complexes based precursor photoresists. Patterns of **d** zinc oxide, **e** copper oxide, and **f** zirconium dioxide manufactured using our strategy (zinc oxide at 25 mW and 2 mm s⁻¹, copper oxide at 20 mW and 3 mm s⁻¹, and zirconium dioxide at 20 mW and 10 mm s⁻¹). Scale bar: 5 μm. SEM images and EDS maps of zinc-containing grid patterns **g** before pyrolysis and **h** after pyrolysis. Scale bar: 50 μm. **i** EDS spectrum taken from an internal beam region of zinc-containing patterns after pyrolysis. **j** TEM selected area electron diffraction (SAED) patterns taken from the pyrolyzed zinc-containing sample. **k, l** High-resolution TEM images obtained from the pyrolyzed zinc-containing sample. Scale bar: 2 nm. **m** XRD spectra of pyrolyzed zinc-containing sample. **n** XRD spectrum of pyrolyzed copper-containing sample. **o** XRD spectra of pyrolyzed zirconium-containing samples. **p** The shrinkage rate of the ZnO patterns after pyrolysis, the inset shows the pattern used in the shrinkage test. Scale bar: 20 μm. The patterns in **f, g,** and **p** were fabricated at 25 mW and 2 mm s⁻¹.

In general, MOS micropatterns will suffer from structural shrinkage due to the removal of organic components during the pyrolysis process. This shrinkage is often random and uncontrollable in the previous works[23,28], which can be fatal to the design and processing of nanodevices. Therefore, it is crucial to predict or quantify the sintering shrinkage of MOS patterns. Since ZnO micropatterns have more application scenarios, we quantitatively investigate its pyrolysis shrinkage. As shown in Supplementary Fig. 8, we fabricate rectangular blocks and parallel lines. It is found that ZnO micropatterns with different line widths have different shrinkage rates. For large rectangular (2.80 μm × 2.85 μm) patterns, the shrinkage rate after sintering is around 6% in both x and y directions. Interestingly, the absolute value of the shrinkage seems to stay between 200 and 300 nm even though the side length of rectangular patterns is increased from 5 to 20 μm. This should result from the anchoring effect of the substrate that resists the in-plane shrinkage (parallel to the substrate) of ZnO micropatterns, and significant shrinkage occurs only in the edge regions that lack sufficient anchoring effect. In view of this, when size decreases, shrinkage is undoubtedly increased dramatically (Fig. 1p). Typically, the shrinkage rate is as high as 16% (433 nm after sintering) for fine parallel lines with a line width of 516 nm (Supplementary Fig. 8). At sizes below 1 μm, the shrinkage no longer seems to be size-dependent and varies in the range of 12–20%. (Supplementary Fig. 9). It is worth noting that shrinkage rates of MOS micropatterns reported in the previous literature on MOS are generally as high as 50–70%[23,28]. In contrast, we offer a strategy that has the distinct advantage of more realistically reproducing the original design structure without the need for extensive compensation. The high metal content in precursor photoresist is responsible for the ability of our strategy to fabricate high-fidelity and low-shrinkage MOS micropatterns. As shown in Supplementary Table 1, the precursor photoresist film is almost entirely composed of acrylic metal complexes (wt% > 98%), which contain metal elements up to 15.05–30.59%. Meanwhile, metal ions will not be washed out during the development process since they are bound to

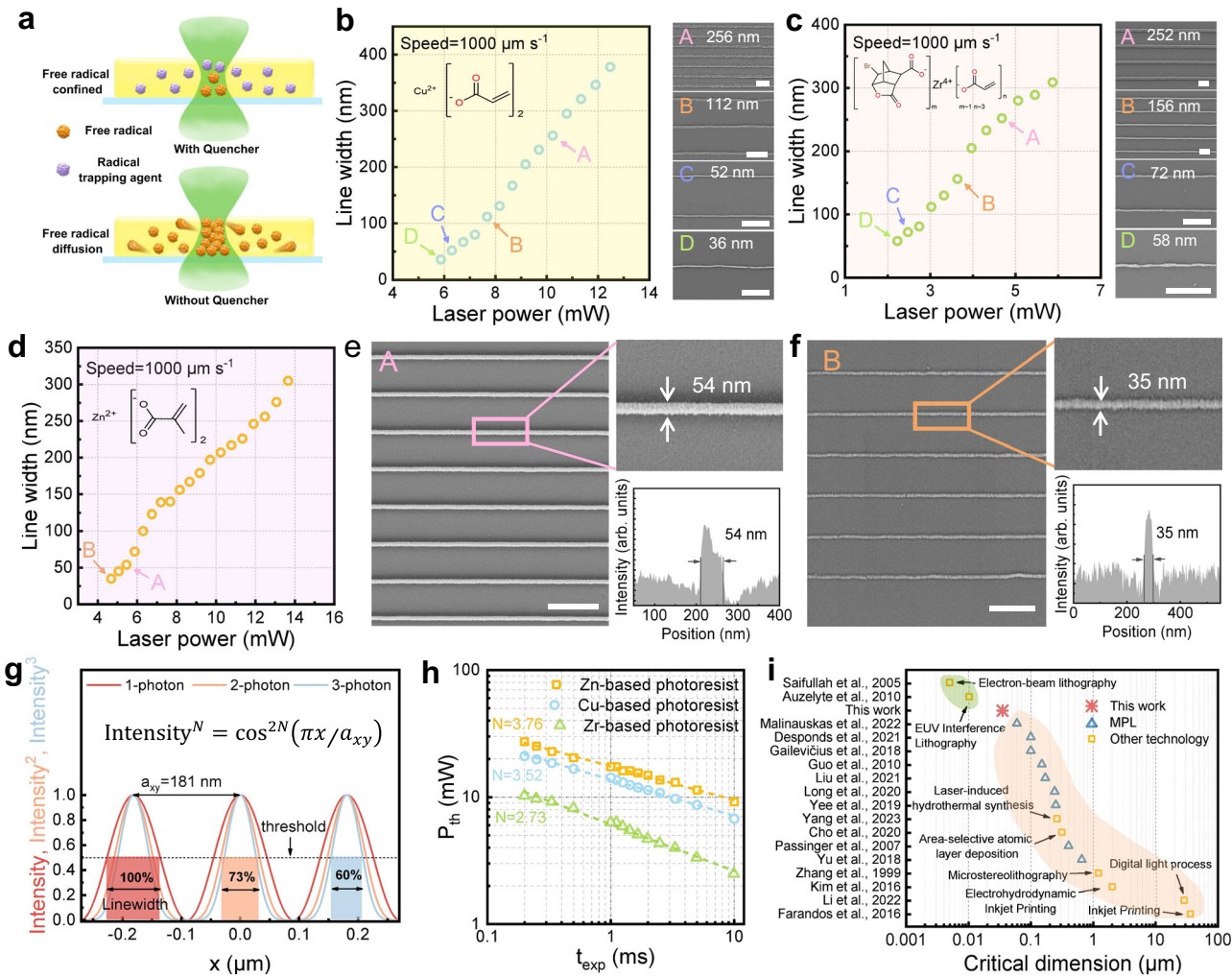

**Fig. 2 | Ultra-high precision nano additive manufacturing of MOS via MPL. a** The diffusion of free radicals can lead to the occurrence of unexpected polymerization reactions. The addition of BTPOS (a radical trapping agent) can inhibit the unwanted polymerization reactions and reduce feature size. The relationship between the line widths and laser power for CuO (**b**), ZrO₂ (**c**), and ZnO (**d–f**). Scale bar: 500 nm. **g** Theoretical linewidth comparison of one-photon, two-photon, and three-photon exposures using parallel lines as models. For a fixed exposure time, the threshold dose is converted to the threshold intensity, and the theoretical line width is obtained from the full width at half maximum (FWHM). *N* represents the nonlinearity absorption exponent of photoresists to the laser. $a_{xy}$ is the critical period of photoresist under specified conditions in MPL. **h** The double-logarithmic representation of the excitation laser power ($P_{th}$) versus the exposure time ($t_{exp}$) for MPL. The dashed curve is a linear fit to the data, and nonlinearity absorption exponent *N* is calculated from the slope of the linear fit line ($N = -1/slope$). **i** Comparison of critical dimensions of MOS additive manufacturing technologies. Data and references are listed in Supplementary Table 2.

the polymer network through coordination bonds. Intersecting microstructures in devices is unavoidable and may lead to more defects due to stress anisotropy during shrinkage. As shown in Supplementary Fig. 10, thanks to the low shrinkage rate of our strategy, the anisotropic shrinkage does not give rise to the fracture and destruction of the grid MOS (ZnO), demonstrating the high fidelity of our strategy.

**Ultra-high precision nano fabrication of metal oxides**
As the integration degree of microelectronic devices continues to increase, the requirements for the CD of MOS have also risen. Although MPL can exceed the diffraction limit and achieve a feature size of nearly 100 nm with the aid of nonlinear absorption, the migration of free radicals hinders the further improvement of precision. In addition to utilizing solid-state precursor photoresists to reduce free radical migration, we further introduce free radical trapping agent (Fig. 1a, BTPOS) into photoresists, whereby free radicals can be confined to a very small space (Fig. 2a), and thus facilitates ultra-high-precision additive manufacturing. As shown in Fig. 2b–d, at a fixed scanning

speed of 1000 μm s⁻¹, the line widths of all three oxides, CuO, ZrO₂, and ZnO, become progressively smaller as the laser power decreases until they cannot be retained in the development process. It should be noted that the scatter data in Fig. 2b–d tend to be more linear than those reported in the literature[44–46], which may be due to the introduction of BTPOS and the sintering shrinkage. Specifically, the CD of CuO lines is 36 nm, and better surface morphology can be obtained at 52 nm (Fig. 2b). For ZrO₂, CD can only reach 58 nm and high-quality lines are achieved at 72 nm (Fig. 2c). Remarkably, high-quality ZnO line arrays are consistently realized at line-width of 54 nm (Fig. 2e) and even 35 nm (Fig. 2f). Apparently, Zn-based acrylic metal complexes enable the highest additive manufacturing precision for ZnO, after all, all precursor photoresists use the same initiator, radical trapping agent, and ratios. However, the underlying principles seem unclear, so we will look into it next.

It is believed that CD depends on the nonlinearity absorption exponent (*N*) of the photoresist, apart from wavelength (λ), numerical aperture (NA), and technical factor (*k*)[47–51]. Wegener's team discussed the resolution of DLW in detail in their work, which we refer to here[52].

According to the usual Fraunhofer diffraction formula of a grating, we can derive the Abbe condition. As shown in Fig. 2g, assume a single parallel exposure to the photoresist, forming a grating with critical period $a_{xy}$. With a laser wavelength of 525 nm and an NA value of 1.45 for the objective lens, we can determine an $a_{xy}$ of 181 nm. To determine the exposure dose, we can assume that the local exposure dose is simply proportional to the density of molecules excited by light absorption. For one-photon absorption, the exposure dose is proportional to the light intensity at a fixed exposure time. For two-photon (three-photon) absorption, the exposure dose is proportional to the square (cube) of the light intensity at a fixed exposure time. It can be obviously seen that the larger the nonlinear absorption exponent is, the smaller the full width at half maximum (FWHM) of the exposure dose is (the minimum achievable line width is about 40% of the FWHM). Put it another way, the larger the $N$, the smaller the CD[53]. We explore the threshold power ($P_{th}$) of the above three photoresists and then plot a linear fit between the $P_{th}$ and exposure time ($t_{exp}$) through a double-logarithmic representation (Fig. 2h). Nonlinearity absorption exponent $N$ is calculated from the slope of the linear fit line ($N = -1/slope$). Clearly, Zn-based precursor photoresist has the highest $N$ (3.76), followed by Cu-based precursor photoresist ($N = 3.52$) and Zr-based precursor photoresist ($N = 2.76$), which leads to the order of their CD (ZnO (35 nm) <CuO (36 nm) <ZrO$_2$ (58 nm)) and is consistent with the above theory. Meanwhile, it also can be seen that Zn-/Cu-based precursor photoresists have strong absorption in the deep UV region (Supplementary Fig. 11), but not for Zr-based precursor photoresists, suggesting Zn-/Cu-based precursor photoresists are more likely to absorb the femtosecond laser through higher-order non-linearities (bigger $N$) to achieve high-precision. It is generally believed that the $N$ of precursor photoresist is normailly determined by both the resin (acrylic metal complexes) and the initiator (DETC), and further dominated by the excitation energy and ionization energy of them[35,54]. From DFT calculations (Supplementary Table 3), Zn-based acrylic metal complexes show larger excitation/ionization energies and more equivalent photon numbers than the others, making it possible to have a bigger $N$. Since the excitation energy and ionization energy rely on the molecular structure of acrylic metal complexes. Therefore, it is expected to obtain photoresists with higher $N$ by changing the central metal ions and ligands of acrylic metal complexes, thus realizing further improvement of CD in the future.

Undeniably, the CD of the final MOS should also be related to the shrinkage of the precursor photoresist during polymerization and sintering. The polymerization shrinkage tends to be proportional to the polymerization crosslink density of the resin. Zr-based acrylic metal complexes contain inactive groups with large steric hindrances and are unable to achieve high cross-linking densities, while polymerization shrinkage will be further inhibited by the alicyclic groups in their ligands. In contrast, both Zn and Cu-based acrylic metal complexes have two active sites with small steric hindrance, making them easier to realize high polymerization crosslink density. In this sense, acrylic metal complexes with more active sites and less steric hindrance are desirable for improving CD. As mentioned above, shrinkage during sintering is mainly due to organic component removal. The ratio of the initial mass of the acrylic metal complexes to the mass of the final resulting MOS can reflect its shrinkage potential. As shown in Supplementary Table 1, the order of mass reduction of the three metal composites before and after pyrolysis is Zr > Zn > Cu. It should be noted that a large mass loss during the pyrolysis is likely to cause more defects and poor quality, and the lines may be broken and cannot be retained during the shrinkage process, so it is not necessarily conducive to the improvement of CD.

Figure 2i and Supplementary Table 3 show the comparison of CD of MOS realized by various additive manufacturing technologies[15,23,25,28,42,55–65]. It is clear that the performance of MPL is much better than the other reported additive manufacturing strategies thanks to the nonlinear absorption, but it still does not exceed 100 nm due to the rapid migration of free radicals in their material system. In contrast, by utilizing solid-state precursor photoresists in combination with free radical trapping agents, we surpass the optical diffraction limit and achieve an astonishing CD of 35 nm. Not only is this the benchmark for MOS additive manufacturing, it also seems to exceed the extreme lithography capabilities of all MPL derived technologies, such as STED lithography[66–69]. In addition, while extreme ultraviolet interference lithography (EUV-IL) and electron beam lithography (EBL) can achieve 10 nm or even 5 nm MOS patterns by virtue of their huge advantages in wavelength, they are difficult to avoid the use of masks or involve complex pattern transfer processes. At the same time, the above two techniques are too expensive for preliminary scientific exploration. Therefore, this work provides a good alternative for certain high-precision MOS patterning needs.

## Atomic doping and heterogeneous metal oxides

In many cases, atomic doping is often used to modulate the bandgap of a semiconductor (known as extrinsic semiconductor) to obtain the target characteristics[70–72]. Therefore, we use ZnO as an example to explore the possibility of Al-doping based on our strategy. Aluminum acrylate is mixed into the photoresist, and the rest of the process remains unchanged. Aluminum acrylate acts as a reactive monomer during the polymerization process and participates in the construction of a robust polymer network (Supplementary Fig. 12a), ensuring that the Al atoms are not lost during the development process. We fabricate an Al-doped ZnO pattern and use a variety of methods to characterize the doping effect. As shown in Supplementary Figs. 12b, c, the EDS mapping results indicate that the sintered pattern mainly contains Zn and O elements, and the Al elements before and after pyrolysis are difficult to recognize due to the low doping amount. Furthermore, the EDS spectrum (Fig. 3a, b) taken from an internal beam section shows that the atomic percentages of Zn, O, and Al are 30.65 at%, 32.29 at%, and 0.87 at%, respectively, resulting in an Al to Zn ratio of 0.028. The ratio of Al atom to Zn atom in raw material is about 0.01, and the main reason for the deviation of the results is the inaccuracy of measuring light elements using EDS. Figure 3c shows the HRTEM images of Al-doped ZnO, the resultant space lattices are 0.285 nm for (100) and 0.265 nm for (002). Compared to the pure ZnO in Fig. 1m, this lattice reduction is clearly due to the doping of Al atoms, which induces a deformation of the lattice[73]. Lattice reduction due to Al-doping can also be confirmed by XRD. As can be seen from Supplementary Fig. 13, the Al-doping has no influence on the number and intensity of diffraction peaks, indicating that no alumina crystals are generated. However, the diffraction peak position of Al-doped ZnO is found to be slightly shifted to a larger diffraction angle (Fig. 3d), suggesting that the Al atoms reduce the lattice constant of ZnO[74]. The lattice constant is expected to be shorter when Al atoms are substituted into the Zn sites since the ionic radius of Al$^{3+}$ (0.53 Å) is smaller than that of Zn$^{2+}$ (0.74 Å)[75]. Therefore, we can conclude that Al has been successfully doped into the ZnO crystal at the atomic level and sand that other doping might be achieved by a similar strategy.

Since many devices are integrated from multiple materials, i.e., heterogeneous structures. Thus, we attempt the fabrication of heterogeneous MOS based on MPL. As shown in Fig. 3e–g, we design a quick response (QR) code pattern that consists of ZnO (yellow part), CuO (blue part), and ZrO$_2$ (green part). By repeating coating-exposure-development three times and subsequent pyrolysis, this heterogeneous QR code is successfully realized (Fig. 3h–j). Due to the difference in refractive index of the three MOS, they exhibit different brightness under the optical microscope. From the SEM image in Supplementary Fig. 14, the clear boundaries of the MOS indicate that the MOS prepared in the previous step do not affect the additive manufacturing of the subsequent MOS. Figure 3k–m shows the EDS spectrums and the corresponding elemental maps of the QR code

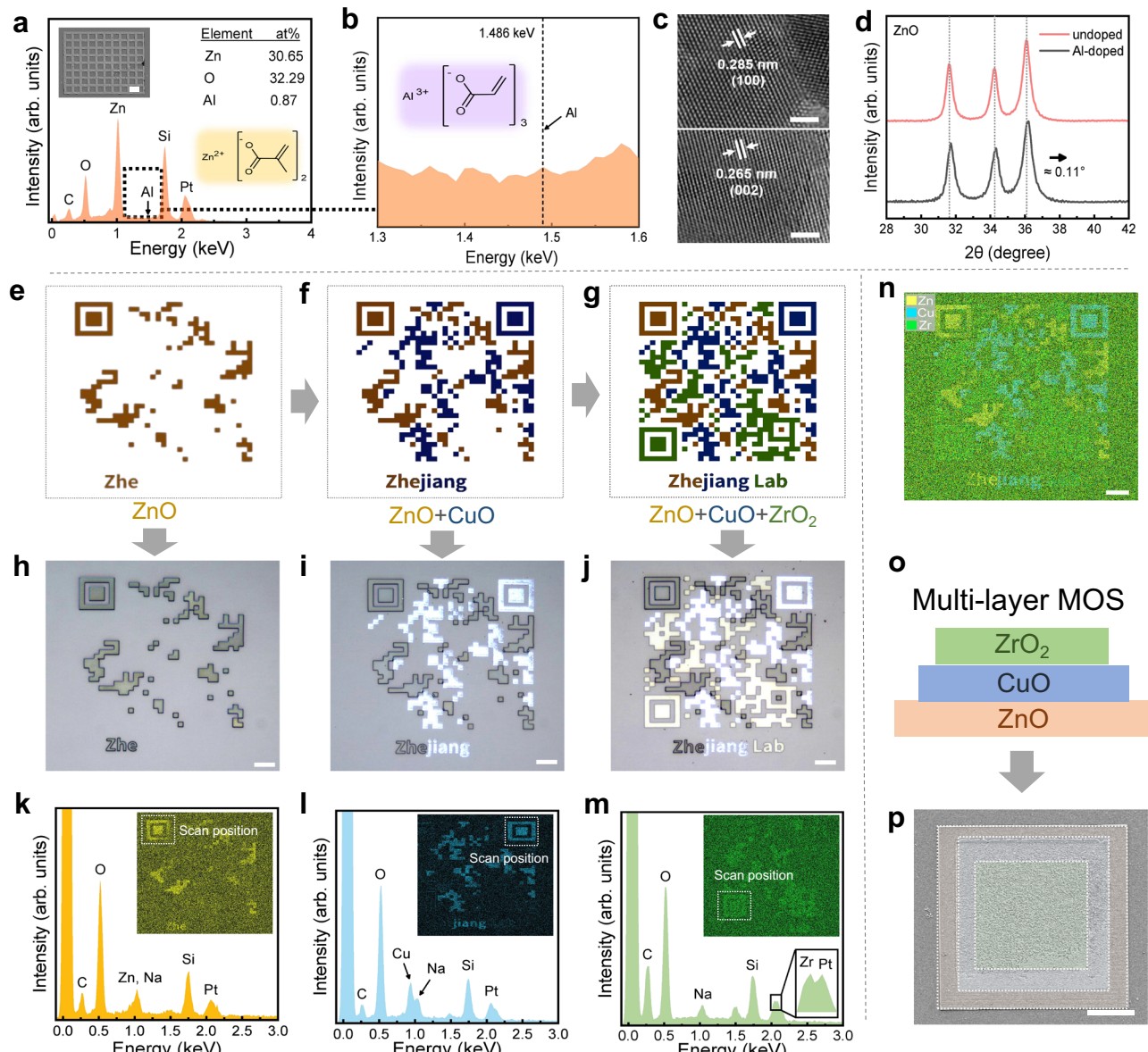

**Fig. 3 | Atomic doping and heterogeneous MOS via MPL. a, b** EDS spectrum taken from an internal beam region of Al-doped zinc-containing patterns after pyrolysis, the patterns were fabricated at 25 mW and 2 mm s⁻¹. Scale bar: 10 μm. **c** High-resolution TEM images obtained from the pyrolyzed Al-doped zinc-containing sample. Scale bar: 2 nm. **d** XRD spectra of pyrolyzed zinc-containing sample and pyrolyzed Al-doped zinc-containing sample. **e–g** Manufacturing schematic diagram of heterogeneous MOS pattern via MPL and **h–j** corresponding optical microscope images (25 mW and 2 mm s⁻¹ for (**h**), 20 mW and 3 mm s⁻¹ for (**i**), and 20 mW and 10 mm s⁻¹ for (**j**). Scale bar: 10 μm. **k–m** EDS spectrum taken from an internal beam region of different positions of the QR code. **n** EDS maps of the QR code pattern. Scale bar: 10 μm. **o, p** Schematic diagram of heterogeneous multi-layer MOS (25 mW and 2 mm s⁻¹ for ZnO, 15 mW and 5 mm s⁻¹ for CuO, and 15 mW and 10 mm s⁻¹ for ZrO₂) and corresponding SEM images. Scale bar: 10 μm.

pattern. Among them, the distribution image of Cu is very clear, and the peak of Cu can be accurately distinguished from the EDS spectrum (Fig. 3l). Although the peaks of Zn (1.012 keV) and Na (1.041 keV, from the substrate) overlap, as do peaks of Zr (2.042 keV) and Pt (2.048 keV, from the conductive layer), resulting in a relatively fuzzy distribution image, the patterns can still be distinguished (Fig. 3k–m). By selective elemental imaging of Zn, Cu, and Zr, a color QR code can be obtained (Fig. 3n). This may provide an alternative strategy for encrypting information at the micro-scale. In addition to the two-dimensional QR code, a three-layer multi-MOS consisting of ZnO, CuO, and ZrO₂ is also successfully implemented (Fig. 3o–p), which means that this strategy has the potential to achieve layer-by-layer fabrication of heterogeneous MOS. Meanwhile, our strategy can also be used to fabricate real 3D micro-structures (Supplementary Fig. 15). In a word, this

demonstrates the feasibility of fabricating heterogeneous MOS by the present strategy, thus fulfilling various requirements for semiconductor devices.

In addition, another significant advantage of the provided solid precursor photoresist is that it is compatible with air objectives for large-scale additive manufacturing (LS-AM). Indeed, LS-AM is the basis for truly widespread applications, a barrier that liquid photoresists have struggled to overcome. Supplementary Fig. 16a–c shows the lines printed using solid precursor photoresists and an air objective (50×, NA = 0.95). Due to the smaller numerical aperture, the CD is increased over the use of oil objectives but remains below 150 nm. Moreover, a deterioration in line edge roughness is also observed. One possible reason here is that the light intensity edge gradient of the focused spot of different NA objectives is different. The intensity gradient of the air

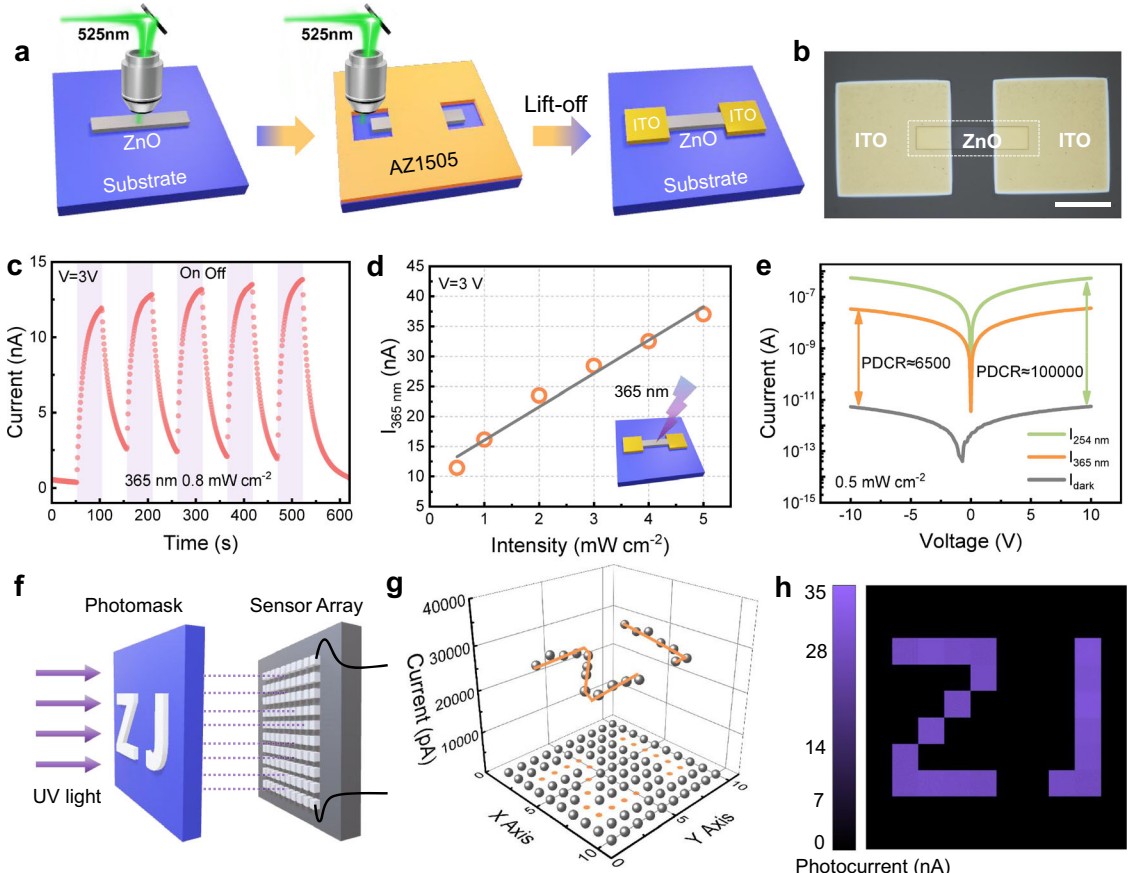

**Fig. 4 | ZnO UV photodetector fabricated by our strategy and the performance improvement. a** Step-by-step schematic diagram of the manufacturing process of ZnO UV photodetector. **b** Optical microscope image of ZnO UV photodetector. Scale bar: 50 μm. **c** Time-dependent photoresponse of ZnO UV photodetector under a pulsed 365 nm UV illumination (0.8 mW cm⁻²) with 3 V external bias.

**d** Current response of ZnO UV photodetector under different intensities of 365 nm UV light (3 V bias). **e** I−V characteristic curves of ZnO UV photodetector in the dark and under 365 nm and 254 nm UV illumination (0.5 mW cm⁻²). **f** Schematic illustration of the UV imaging. **g** Current distribution during imaging and (h) imaging results.

objective (NA = 0.95) is smaller than that of the oil objective (NA = 1.45), so the edge roughness of the exposure line worsens. Another reason is the standing wave effect, which will not only cause a decrease in adhesion of the line edge but also make a difference in the developing environment at the edge of the line. Some small monomers in photoresist are more fully developed than structures attached to the substrate. Notably, the Zr-based photoresist showed the highest precision, possibly because of its better adhesion to the silicon surface. Subsequently, a Zr-based photoresist is used for LS-AM, and a 2 cm × 2 cm grating with a line width of 190 nm and a resolution of 500 nm is achieved on silicon (Supplementary Fig. 16d, e). It should be noted that the performance of LS-AM is closely related to the stability of the MPL system, including the stability of the laser power and the consistency of the focal plane position. We only make a simple demonstration here to illustrate the applicability of the solid precursor photoresist we developed for LS-AM. There are still many technical issues in practical application, as well as improvements to the formulation of photoresist.

## Photodetectors fabrication and performance improvement

ZnO is of great value in UV sensing with a wide direct band-gap (3.37 eV) and large exciton binding energy (≈60 meV) at room temperature[76,77], and its UV sensing mechanism is depicted in Supplementary Figs. 17 and 18. We manufactured some UV photodetectors based on pure ZnO and Al-doped ZnO through our strategy. A schematic diagram of the fabrication process of the UV photodetectors is given in Fig. 4a, and the as-prepared photodetector unit is shown in Fig. 4b. Under 365 nm irradiation

(0.8 mW cm⁻²), the decay time (the time interval between 90% and 10% of the maximum current value) of ZnO UV photodetector is 62 s with good photo-cycle stability (Fig. 4c), and the photo-to-dark current ratio (PDCR) is up to 10,500 at an external bias of 3.0 V (Supplementary Fig. 19a). In case of Al-doped ZnO UV photodetector, the decay time is reduced to 24 s and the PDCR is significantly increased to 16,000 which proves the effectiveness of Al doping (Supplementary Fig. 19b, c). The performance improvement after Al doping is mainly attributed to the generation of more excited electrons and the edge effect[78,79]. Meanwhile, the response current of the ZnO photodetector increases linearly with irradiation intensity for both 365 nm and 254 nm (Fig. 4d and Supplementary Fig. 19d), which means that the fabricated ZnO UV photodetector can be used to measure the UV intensity of a certain wavelength after calibration. Besides, the fabricated ZnO UV photodetector seems to be more sensitive to 254 nm, e.g., a photo-to-dark current ratio of up to approx 100,000 (Fig. 4e), which greatly exceeds that of 365 nm (6,500). The photon energy of 365 nm UV light is 3.40 eV, which is close to the bandgap of ZnO (3.37 eV). Due to the non-zero spectrum width, there still exist photons with energies below the band gap, which cannot generate electron−hole pairs. The photon energy of 254 nm ultraviolet light is 4.88 eV, which is much higher than the band gap of zinc oxide, and can easily excite electrons to the conduction band, which is why the ZnO UV detector is more sensitive to 254 nm ultraviolet light. Notably, the superior performance of our devices can be clearly demonstrated by comparison with similar literature (Supplementary Table 4). Meanwhile, the

PDCR of the ZnO UV photodetector might be further improved by our strategy to reduce the line width of ZnO (Supplementary Fig. 20).

Subsequently, a $10 \times 10$ array of ZnO photodetector is further fabricated on a 1-in. oxidized wafer (Supplementary Fig. 21) to evaluate its potential for UV imaging. The imaging system is illustrated in Fig. 4f, the ZnO UV photodetector array acts as the sensing units, and a photo-mask with the hollowed pattern ZJ is used to block the light source (365 nm), where the response current of ZnO UV photodetector units is measured at 5.0 V bias. As shown in Fig. 4g, the response current values between the exposed unit and the shaded unit differ significantly, resulting in a high imaging contrast (Fig. 4h) which coincides with the hollowed pattern ZJ on the photo-mask. Therefore, it can be concluded that the ZnO photodetectors obtained by the presented additive manufacturing strategy have excellent UV sensing and imaging capabilities. Except for UV sensing, the ZnO semiconductor has many other applications in microelectronics, such as thin-film transistors[80], light-emitting devices[81], and gas sensing devices[82]. In addition, CuO can be used in gas sensors[83] and biosensors[84], and $ZrO_2$ can be employed in ferroelectric devices[85] and piezoelectric devices[86]. Noted that many other components on the target device may not be compatible with the sintering temperature of MOS, one alternative is to print components and then assemble them. In conclusion, the proposed strategy for high-precision additive manufacturing of MOS patterns has enormous potential in the development and manufacture of microdevices.

## Discussion

Here, we report a strategy using metal-organic compounds as solid precursor photoresists for the MPL, to realize ultra-high precision nano additive manufacturing of metal oxide semiconductors. Through MPL and post-pyrolysis, metal oxide patterns, including ZnO, CuO, and ZrO2, can be obtained successfully. Thanks to the limitation of radical migration by solid-state photoresists and the quenching effect of radical trapping agents, the CD of all three metal oxide patterns are below 60 nm, with ZnO reaching a CD of 35 nm, which sets a benchmark for nano additive manufacturing of metal oxides. In addition, we successfully realize atomic doping in metal oxide patterns by mixing trace elements in solid precursor photoresists, which could be used to modulate the bandgap of a semiconductor to obtain the target characteristics. Meanwhile, 2D and 3D heterogeneous structures that consist of ZnO, CuO, and $ZrO_2$ can also be created by multiple MPL, allowing this strategy to accommodate the various requirements of semiconductor devices. Finally, we fabricate some UV photodetectors by MPL based on the pure ZnO and Al-doped ZnO. It is reasonable to speculate that this strategy may be applicable to the manufacture of other metal oxides. Overall, this work may provide an alternative strategy for nano additive manufacturing of highly integrated chips and devices.

## Methods

### Materials

Zinc methacrylate, Aluminum acrylate, propylene glycol 1-monomethyl ether 2-acetate (PGMEA), 1-methoxy-2-propanol, and isopropanol (IPA) are purchased from Adamas. Copper (II) acrylate is purchased from Alfa Aesar. Zirconium bromonorbornanelactone carboxylate triacrylate (PRM30) was purchased from Sigma-Aldrich. 7-Diethylamino-3-thenoylcoumarin (DETC) is purchased from J&K Scientific. Bis (2,2,6,6-tetramethyl-4-piperidyl-1-oxyl) sebacate (BTPOS) is purchased from TCI. Commercial photoresist AZ1505 and MIF developer are supplied by Tansoole. All chemicals are used without further treatment.

### Preparation of metal-based precursor photoresist

For zinc-based precursor photoresist, zinc methacrylate (150 mg) and DETC (3 mg) are dissolved in PGMEA (3.0 g) and then filtered twice through a 0.22 μm membrane to obtain a clear yellow solution. Then,

BTPOS (4.8 mg) was added to the solution and stirred for 60 min to obtain the zinc-based precursor photoresist. For copper-based precursor photoresist, copper (II) acrylate (250 mg) and DETC (5 mg) are dissolved in PGMEA (5.0 g) and then filtered twice through a 0.22 μm membrane to obtain a blue clarifying solution. Then, BTPOS (8 mg) was added to the solution and stirred for 60 min to obtain the copper-based precursor photoresist. For zirconium-based precursor photoresist, zirconium bromonorbornanelactone carboxylate triacrylate (250 mg) and DETC (5 mg) are dissolved in PGMEA (5.0 g) and then filtered twice through a 0.22 μm membrane to obtain a yellow clarifying solution. Then, BTPOS (8 mg) was added to the solution and stirred for 60 min to obtain the zirconium-based photoresist. For aluminum-doped zinc-based precursor photoresist, dissolving zinc methacrylate (148.5 mg), aluminum acrylate (1.5 mg), and DETC (3 mg) into PGMEA (3.0 g), then filtering twice with 0.22 μm membrane to obtain a yellow clarifying solution. Then, BTPOS (4.8 mg) was added to the solution and stirred for 60 min to obtain the Al-doped zinc-based precursor photoresist. For 3D fabrication, the content of metal-organic compounds and initiators in photoresists both increase threefold, while the rest remains unchanged.

### Multi-photon lithography and development

For 2D MOS fabrication, three precursor photoresist films (approx. 30 nm in thickness) are achieved by spin-coating. Specifically, 2 drops of photoresist solution are dropped onto a glass substrate (thickness = 0.17 mm), accelerating the rotary coater to 500 rpm for 10 s, and then accelerating to 2000 rpm for 60 s. For zirconium-based precursor photoresists, a quartz substrate is used to replace the glass substrate since the pyrolysis temperature is up to 1000 °C, which exceeds the maximum temperature that glass can tolerate. MPL is performed using a self-built direct laser writing system, which consists of a femtosecond laser, acousto-optic modulator, beam expander, galvanometer scanner, scan lens, tube lens, objective, mirrors, and piezoelectric stage. A 525 nm femtosecond laser (pulse width: 150 fs, repetition rate: 80 MHz) is used as an irradiation source, and a 100× oil-immersion objective lens (NA = 1.45) is used to focus the laser beam. For large-scale additive manufacturing on wafers, an air objective (50×, NA = 0.95) is used. For micro 3D MOS fabrication, the as-prepared photoresist is drop-casted onto a glass substrate and then dried at 50 °C for 2 h. The laser power used is in the range of 2–30 mW, and the scanning speed is varied from 50 μm s$^{-1}$ to 10,000 μm s$^{-1}$. After MPL, metal-based precursor photoresists are developed with PGMEA for 30 s and then rinsed with IPA for 60 s.

### Pyrolysis

The samples are pyrolyzed in a muffle furnace (MTI KSL-1200X) at ambient pressure in the air at 10 °C min$^{-1}$ to a set temperature (ZnO and CuO at 550 °C, and $ZrO_2$ at 550 °C, 800 °C, and 1100 °C, respectively, to explore the transformation of its crystalline phases) for 60 min and then cooled naturally to room temperature.

### Fabrication and characterization of ZnO photodetector unit

The Zn-based precursor photoresist is spin-coated (500 rpm for 10 s, followed by 2000 rpm for 60 s) onto the surface-oxidized silicon (with 300 nm $SO_2$ layer). Afterward, a rectangular structure (100 μm × 20 μm) is exposed using MPL (20× air objective) at 25 mW and 2 m s$^{-1}$. After exposure, the sample is developed in PGMEA for 30 s and cleaned with IPA for 60 s to remove the unexposed photoresist. Then, the rectangular ZnO samples can be obtained by sintering at 550 °C for 1 h. ITO (length, width, and thickness are 100 μm, 100 μm, and 100 nm) electrodes are manufactured by MPL and lift-off process. In detail, the commercial photoresist AZ1505 is spin-coated (2000 rpm for 60 s) onto the above substrate and pre-baked at 90 °C for 60 s. Two squares (100 μm × 100 μm) are exposed at each end of the rectangular ZnO using the alignment technique. The spacing of the two squares is

40 μm. Then, post-baking AZ1505 at 90 °C for 60 s, followed by developing with MIF developer (60 s) and water (60 s) to remove the exposed photoresist. ITO electrodes are achieved by sputtering (Denton Vacuum Discovery 635). Finally, the unexposed AZ1505 photoresist is dissolved using acetone to obtain the ZnO UV photodetector unit. The photoelectronic properties are tested by a probe station and a semiconductor parameter analyzer (Agilent B1500A).

## Fabrication and characterization of ZnO imaging array

The preparation method and process parameters of ZnO UV imaging arrays are almost identical to those of the ZnO UV photodetector unit mentioned above. The difference is that the imaging array has 100 photodetector units (10 rows and 10 columns, with a transverse distance of 0.8 mm and a longitudinal spacing of 1.8 mm between the units). The electrodes of the sensing units are led to the edge of the silicon by ITO wires to facilitate probe testing. ITO wires (10 μm in width) are fabricated by the same method with ITO electrodes. The imaging performance is tested by a probe station and a semiconductor parameter analyzer (Agilent B1500A).

## Material characterization

Samples are observed by a field emission SEM (Zeiss Sigma 300) at 1 kV. EDS maps are generated by a Zeiss Sigma 300 equipped with an Oxford X-Max SDD EDS system at 5 kV. TEM images are obtained by a high-resolution TEM (JEOL JEM-2100 Plus). All measurements are performed via ImageJ software. The XRD data is collected by a Rigaku miniflex 600 at 40 kV and 15 mA. The diffractograms are performed from 20° to 80° as the structural information is mainly located in this range. Due to the limitation of sample size, XRD tests are performed on powder materials. The XRD sample is prepared as follows: the photoresist film is polymerized under 365 nm UV light, then a small amount of powder can be obtained by scraping the film off the substrate, and the process is repeated until enough sample is obtained.

## Reporting summary

Further information on research design is available in the Nature Portfolio Reporting Summary linked to this article.

## Data availability

All data are presented in the Article and the Supplementary Information. The data are available from the authors on request. Source data are provided with this paper.

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

## Acknowledgements

This work was supported by the National Natural Science Foundation of China (62125504 to C.K., 22105180 to C.C.), National Key R&D Program of China (2021YFF0502700 to C.K.), "Pioneer" and "Leading Goose" R&D Program of Zhejiang Province (2023C01186 to C.C.), Major Program of Natural Science Foundation of Zhejiang Province (LD21F050002 to C.K.), and Zhejiang Provincial Ten Thousand Plan for Young Top Talents (2020R52001 to C.K.).

## Author contributions

C.C. Investigation, supervision, conceptualization, methodology, funding acquisition, writing—original draft, writing—review & editing. X.X. Investigation, data curation, formal analysis, methodology, writing—original draft. X.S. Investigation, data curation, formal analysis. X.W. Investigation. C.D. and D.Z. Optical systems. Z.Y. Software. Q.L., C.K., and X.L. Supervision, funding acquisition, resources.

## Competing interests

The authors declare no competing interests.
