## [Transparent Peer Review file · Nature Communications]

Ultra-high Precision Nano Additive Manufacturing of Metal Oxide Semiconductors Via Multi-photon Lithography

Corresponding Author: Professor Chun Cao

Version 0:

Reviewer comments:

Reviewer #1

(Remarks to the Author)

In this manuscript, the author presents a novel strategy for achieving ultra-high precision nano additive manufacturing of metal oxide semiconductors (MOS) using solid precursor photoresists based on metal-organic compounds. The traditional multi-step lithography and transfer process for nanopatterning MOS is time-consuming and costly, prompting the development of this new approach. By utilizing metal-organic compounds as solid precursor photoresists for multi-photon lithography (MPL) and post-sintering, the researchers successfully fabricated MOS patterns with critical dimensions as small as 35 nm, setting a new benchmark for nano additive manufacturing of MOS. Key findings include the ability to easily achieve atomic doping, create 2D and 3D heterogeneous structures, and fabricate functional devices like a ZnO photodetector.

I believe this paper will be of interest to the readers of Nature Communications. As a consequence, I recommend the acceptance of the present work after major revision, some issues should be considered to make the manuscript more informative and stronger.

1. Regarding the shrinkage evaluation experiment, Figure S8 shows a rectangular block structure to measure the shrinkage rate, which is not rigorous. The degree of structural shrinkage is closely related to the height of the block. The greater the height, the shrinkage at the top reflects the final shrinkage rate. When the height is low enough, such as a thin sheet, the structure shrinkage will be pulled by the substrate, so the measured shrinkage will be less than the actual value.
2. The relationship between polymer line width and laser power shown in Figure 2 (b, c, d) does not seem to be consistent with the intrinsic relationship of multi-photon polymerization processing. A large amount of literature shows that this relationship should be a non-linear changing relationship, especially near the threshold, this non-linear dependence should be more obvious. The author uses a linear function to fit the relationship between line width and power, which is obviously inappropriate and lacks physical connotation. Of course, it is also possible that there are some additional mechanisms during the photopolymerization process that cause the linewidth-power nonlinearity to be insignificant, so it is recommended that the author delete the fitted straight line and only retain the scatter data.
3. Figure S13a-c shows the deterioration of the edge roughness of the processed nanowires. The first reason explained by the author "may be due to the poor quality of the spot generated by the air objective lens." is not very reasonable. "Poor spot quality" generally refers to deformation of the beam shape or intensity distribution. One possible reason here is that the light intensity edge gradient of the focused spot of different NA objectives is different. The intensity gradient of the air lens 0.95NA is smaller than that of the oil immersion objective, so the edge roughness of the exposure line worsens. Another reason is the standing wave effect proposed by the author. The standing wave will cause two phenomena. One is the decrease in adhesion of the polymer edge (the author has already mentioned), and the other is the difference in the development environment at the edge of the line. The standing wave effect causes some small monomers are more fully developed than structures attached to the substrate.
4. The contrast of the SEM photo in Figure 1(p) is so low that it is almost invisible. Please deal with it.
5. In addition to the transmittance curves of the three photoresists, can the absorbance curve of zinc-based, copper-based and zirconium-based photoresist be added?
6. The author compared various MOS patterning preparation processes developed by other researchers in Table S3. It

seems that the biggest contribution is the author's realization of MOS patterning below sub-100 nanometers. However, the authors did not demonstrate the improvement effect of this high-precision pattern on device preparation and performance. From the perspective of MOS device preparation alone, other methods (listed by the author in Table S3) can also be processed and do not require high-precision preparation methods. Therefore, it would be more breakthrough if the author could demonstrate the advantages of the proposed processing method in device preparation.

Reviewer #2

(Remarks to the Author)

The submitted manuscript "Ultra-high Precision Nano Additive Manufacturing of Metal Oxide Semiconductor" by Chun Cao et al. describes a lithography method using multiphoton polymerization followed by a pyrolysis step to print narrow lines of metal oxide materials. Notably, the authors use a spincoating photoresist strategy rather than a drop casting method in an attempt to reduce the mobility of radicals formed during printing. After exposure with a typical multiphoton lithography printing scheme, the printed structures are pyrolyzed under high temperature to remove organic components leaving a metal oxide semiconducting material. The pyrolysis step results in shrinkage on the order of 20% leading to linewidths of oxide material of range 35 – 72 nm. The authors also present one doped metal oxide as well as use a printed oxide for UV photodetection. While the authors present some stacked layers of materials, they do not show any truly 3D structures typically associated with the multiphoton lithography process. It is not clear from what is presented what is limiting this process from showing a truly 3D structure. While the authors show an impressive linewidth capability (critical dimension, CD) of laser printed oxide material they do not explicitly demonstrate an example of the advantage of this fine feature size. All examples of devices are printed with dimensions that can be achieved with previously demonstrated methods in the literature (see references from the Figure 2i). Overall, the paper is organized well and easy to follow. The results described here may have some interest to the community. I have outlined below points and concerns that the authors should address if this article is to be considered for publication.

- The abbreviations DUV and EUV are used before defining them.
- The authors seem to use the term critical dimension (CD) for both the width of a polymerized line before pyrolysis and after pyrolysis. This becomes confusing, especially when trying to understand Figure 1p. The authors should better differentiate between the measurements taken during these two different steps of their whole process.
- The insets of Figure 1p are very hard to see. These may be better as their own figure, maybe in the supplementary info.
- It seems odd to present the atomic ratios of only select atoms such that they add up to 100%. Is this typically how it is done? It seems to me that you can still show the ratio of the atoms you are interested in without excluding other atoms from the measurement.
- In figure 1h, it seems to me that there is still a little bit of carbon signal in the shape of the grid of printed material. Can the authors better specify the purity of the after pyrolysis sample?
- The authors state "When the CD is small enough, it can be regarded as a three-dimensional shrinkage, which can be up to 20%". Do the authors do any measurements of the vertical/height shrinkage? I don't see any in the manuscript. Then this statement is confusing.
- I did not find the height of the spin-coated material in the manuscript. What is the thickness?
- Did the authors prebake the spincoated substrate before exposure?
- How are the samples developed before pyrolysis?
- The authors in multiple places explain the effect of BTPOS as a scavenger of radicals in their system. Have the authors actually proved that BTPOS behaves this way in their system and has an effect on the CD?
- In the main text, the authors state that the photoinitiator concentration is the same for all photoresists. However, this does not seem to be the case when looking at the amounts described in the Methods section of the paper. Please address this. If the concentrations are not the same, it makes much of the discussion comparing CD of the different materials irrelevant.
- Figures 2g and 2h are introduced out of order in the text. Consider rearranging the panels.
- The authors discuss DFT calculations of the metal complexes (Table S1) and link the excitation energies to the observed higher N values in their resists. Is it then supposed to be understood that the absorption of the metal complexes is significant compared to the photoinitiator? Why do the authors bother using a photoinitiator if the resist absorbs without it? It seems to me that an argument about diffusion would make more sense here, especially considering the papers that the authors cite for this argument (47, 65).
- The ordering of panels in Figure 3 is hard to follow. a-e are read left from right, but then f-p switch to up and down. Please be consistent within the figure.
- The section title "An attempt to develop ZnO UV photodetector" is strange. It seems like the authors succeeded so why call it an "attempt"?
- It seems strange to call Figure 3o&p a 3D structure. In the context of multiphoton lithography, this would only be 2.5D. Do the authors try to print truly 3D structures with maybe only one resist as multiphoton lithography should be capable of this?
- The authors only provide a very broad range of printing powers and speeds. The authors should be more specific and include the conditions of printing for each of the presented results.
- The pyrolysis temperatures used in this work are very large. Many other components of electrical circuits may not be compatible with these temperatures. Can the authors comment on this?
- The authors compare their results to the optical diffraction limit. However, this is confusing because they compare after pyrolysis results to techniques that do not involve any separate step after exposure and development. Their comparison would make more sense if they compare the dimension of their prints before pyrolysis. I am dubious that they beat the STED technique before pyrolysis. To be clear, I am not saying that the authors shouldn't compare their final results to the final results for other techniques. But it seems suspect to frame their argument in comparison to techniques that do not do post-processing steps. Besides, couldn't STED be applied to their resist system leading to even smaller CD (considering the photoinitiator is DETC)?

- The labeling of scalebars in many of the images is missing.
- What is the spacing of the ZnO UV imaging array? Should provide more info than just 10 x 10.

Reviewer #3

(Remarks to the Author)

The research manuscript is valuable and timely, has significant advancements in the field, but the description is too generic, pompous, and vague due to lack of support.

TITLE:

What is “ultra-high precision”? – it’s a vague and not supported.

What is “metal oxides semiconductor?” – plural to singular material/device?

MPL, which is the employed technique could be added to the title making it more informative of what technique was used instead of hollow calling it “ultra-high precision”.

“Typically, we fabricated ZnO photodetector..” – typically fabricated? What does it mean. And in the main text it is explained as an attempt to fabricate, while in the Discussion it converts to fabricated photodetectors. So was it one, many, or just an attempt to make it?

ABSTRACT

Why specifically 35 nm are referred as critical dimensions (CD)? It is not supported later in the manuscript either.

INTRODUCTION:

CD are mentioned to be of struggle reaching below 80-100 nm, yet have a look at recent related publication on that matter, which also employs solid MOS photoresist and achieves 60 nm: Laser additive manufacturing of Si/ZrO₂ tunable crystalline phase 3D nanostructures. *Opto-Electron Adv* 5, 210077 (2022); <https://doi.org/10.29026/oea.2022.210077>

“Atomically economical..”- what does it stand for?

The mentioned limited Library of photoresists for MPL combined with pyrolysis is actually has been provided in a recent perspective paper at Fabrication of Glass-Ceramic 3D Micro-Optics by Combining Laser Lithography and Calcination, *Adv. Func. Matter.*, 2215230 (2023); <https://doi.org/10.1002/adfm.202215230>.

Among Bauer and Xiong, there was also Gonzalez, who fabricated crystalline micro-optics using very similar photoresist, exposure and post-processing strategy, should be included in overview: Laser 3D Printing of Inorganic Free-Form Micro-Optics, *Photonics* 8, 577 (2021); <https://doi.org/10.3390/photonics8120577>

RESULTS:

“Green laser..” – was it the color of the laser packaging box? Anyhow, “green” lasers have been successfully exploited to fabricate via MPL means even not photo-sensitized resins, see: Femtosecond-Laser Direct Writing 3D Micro-/Nano-Lithography Using VIS-Light Oscillator, *J. Centr. South Univ.*, 29, 3270-3276 (2022); doi: 10.1007/s11771-022-5153-z.

There is not 3D structure demonstrated, though it is mentioned as the MPL is advantageous for it. Why? Previous pointed publications available at <https://doi.org/10.29026/oea.2022.210077> and <https://doi.org/10.3390/photonics8120577> clearly demonstrate the feasibility of the method to construct 3D architectures of complex geometries and high resolution.

What are the photo structuring mechanisms? It is shown in some details, but gives an ambiguous impression. Fig 2 (b, c, and d) shows line width dependence on laser power (P), in (g) it is plotted at a log-log scale and in (h) explained as intensity^N (I^N) law, but in the caption it is mentioned as threshold dose (D or E). So which is the determining parameter and how they are (inter-)related? There is a recent paper studying mechanisms for various wavelengths and pulse durations, revealing the mechanisms contributing to voxel growth, should be discussed and compared: X-photon laser direct write 3D nanolithography, *Virt. Phys. Prototyp.* 18, e2228324 (2023); <https://doi.org/10.1080/17452759.2023.2228324>

There is no image of the real fabricated photodetector in Fig. 4. And it is not clear of what dimensions were used and if the critical 35 nm was necessary or beneficial to achieve it.

DISCUSSION

“new strategy” appears not to be new anymore once the pointed reports will be referenced, but rather an improvement of the previous works, which is substantial.

REFERENCES

At least [2,4,7,8] have no page numbers.

SUPPLEMENTARY

Table S3 should be updated with corresponding paper in <https://doi.org/10.29026/oea.2022.210077> benchmarking 60 nm linewidth in 3D nano-structure.

Version 1:

Reviewer comments:

Reviewer #1

(Remarks to the Author)

The author has already solved the problems I raised in the review. The manuscript can be considered for publication. Some prospects: 1. The effect of demonstrating the advantages of 35 nm extreme lithography resolution and device performance is still not good. I hope the author can find better devices to systemize this feature in the future. 2. The author proposed an additive manufacturing technology for printing semiconductor oxides. Some semiconductor and photonic devices with true 3D architecture should be selected to demonstrate the advantages of this method. Obviously, the manuscript lacks data.

Reviewer #2

(Remarks to the Author)

In my opinion the authors have sufficiently addressed the reviewer comments. The manuscript can be accepted for publication. I have just some minor comments.

- 1) I suggest the authors to mention the film thickness earlier in the main text. This is a VERY thin film being used and most of the multiphoton lithography literature cited in the paper does not use anything close to that thin of a resist coating.
- 2) It is not clear what Supplementary figure 15a is supposed to be. The authors call it an array of nuts. Do they mean bolts?

Reviewer #3

(Remarks to the Author)

Most of the expressed remarks were addressed completely making the manuscript suitable for the publication in its current form.

One more note, Fig 2 (i) remains not updated with the current reference [32] benchmarking 60 nm achievement of year 2022. The inclusion of it was made in the Supplementary Table 2, but not here.

This would make the inset picture complete indicating the consistent progress and most recent advancements in the rapidly evolving field.

Thus I recommend updating it. Sorry, for not noticing it and pointing directly from the first round of review.

Point-to-point Response to Reviewers

Reviewer #1 (Remarks to the Author):

In this manuscript, the author presents a novel strategy for achieving ultra-high precision nano additive manufacturing of metal oxide semiconductors (MOS) using solid precursor photoresists based on metal-organic compounds. The traditional multi-step lithography and transfer process for nanopatterning MOS is time-consuming and costly, prompting the development of this new approach. By utilizing metal-organic compounds as solid precursor photoresists for multi-photon lithography (MPL) and post-sintering, the researchers successfully fabricated MOS patterns with critical dimensions as small as 35 nm, setting a new benchmark for nano additive manufacturing of MOS. Key findings include the ability to easily achieve atomic doping, create 2D and 3D heterogeneous structures, and fabricate functional devices like a ZnO photodetector. I believe this paper will be of interest to the readers of Nature Communications. As a consequence, I recommend the acceptance of the present work after major revision, some issues should be considered to make the manuscript more informative and stronger.

Response:

We appreciate your positive comments and constructive suggestions, which certainly help us to improve the quality of our manuscript. To address your concerns, we have carefully considered your comments and carried out additional experiments to make our manuscript more convincing.

1. Regarding the shrinkage evaluation experiment, Figure S8 shows a rectangular block structure to measure the shrinkage rate, which is not rigorous. The degree of structural shrinkage is closely related to the height of the block. The greater the height, the shrinkage at the top reflects the final shrinkage rate. When the height is low enough, such as a thin sheet, the structure shrinkage will be pulled by the substrate, so the measured shrinkage will be less than the actual value.

Response:

We appreciate the comments, you raised a very good question. As mentioned by you, there is no doubt that “the degree of structural shrinkage is closely related to the height of the block”. Recently, we have reported two kinds of MPL photoresists, and studied their shrinkage rate using a cube with a height of 25 μm [Ref.1-2]. We also found that the shrinkage of the part in contact with the substrate is somewhat limited by the substrate, so we use a height of 25 μm to achieve the real shrinkage rate of the obtained micro 3D structures (Fig.R1).

Fig. R1 The cubes with a height of 25 μm have been used to measure the shrinkage rate of the obtained micro 3D structures. [Ref.1-2]

However, we think this cannot negate the credibility of the results described in **Supplementary Fig. 8-9** (as well as **Fig. 1p**), which were specifically designed for the high-precision ROS fabrication. It is worth noting that in all the 2D MPL experiments (including rectangular block structure and lines), the thickness of all the photoresist films is the same, so theoretically they are similarly constrained by the substrate. Therefore, the test results of **Supplementary Fig. S8-9** (as well as **Fig. 1p**) are plausible, applicable, and meaningful for ultra-high-precision ROS nanofabrication.

Simply put, we provided an “**in-situ**” test method that truly reflects the shrinkage of the high-precision ROS lines. If we switch to a high cube, it is instead an “**ex-situ**” method and the error will be much larger.

In conventional semiconductor lithography (EUV/EBL), the thickness of the photoresist film tends to be thinner, typically in the tens of nanometers, in order to obtain a satisfactory pitch (half-pitch is equivalent to line-width). This is why we obtained metal-organic compound films by spin-coating for multi-photon lithography. It is also why we use thin films to test their shrinkage.

In fact, shrinkage is not only limited by the substrate, but is also closely related to the composition of the photoresist, the dose of exposure (degree of polymerization), the manner of sintering (rate of warming, temperature of sintering), the type of substrate, and the surface properties. Providing a comprehensive picture of shrinkage under arbitrary parameters is impossible to accomplish, and requires the reader to simply explore as needed.

Ref.

1. Cao, C., et al., Dip-In Photoresist for Photoinhibited Two-Photon Lithography to Realize High-Precision Direct Laser Writing on Wafer. *ACS Applied Materials & Interfaces*, 2022. 14(27): 31332-31342.

2. Cao, C., et al., Polyvinylpyrrolidone Hybrid Photoresist for Two-color Sensitive Direct Laser Writing. *Acta Polymeric Sinica*, 2022. 53(6): 608-616.

2. The relationship between polymer line width and laser power shown in Figure 2 (b, c, d) does not seem to be consistent with the intrinsic relationship of multi-photon polymerization processing. A large amount of literature shows that this relationship should be a non-linear changing relationship, especially near the threshold, this non-linear dependence should be more obvious. The author uses a linear function to fit the relationship between line width and power, which is obviously inappropriate and lacks physical connotation. Of course, it is also possible that there are some additional mechanisms during the photopolymerization process that cause the linewidth-power nonlinearity to be insignificant, so it is recommended that the author delete the fitted straight line and only retain the scatter data

Response:

We appreciate the comments very much. We neglected to mention this earlier; linear fitting is indeed meaningless. We have deleted the fitted straight lines in the revised manuscript (**Fig. 2b-d**).

In MPL, the nonlinear absorption of the femtosecond laser by the photoresist results in a nonlinear relationship between line-width and exposure dose as well. This has been extensively studied and reported [**Ref.1-4**], including some of our previous work [**Ref.1-2**].

However, it is also true that the results in **Fig. 2b-d** tend to be more linear than those reported in the literature, which may be due to two reasons: 1. It is due to the introduction of a polymerization blocker (BTPOS) in our photoresist, which can bind to the free radicals and terminate the polymerization reaction, thus weakening the linewidth-power nonlinearity. 2. It is caused by the sintering shrinkage, the results in **Fig. 2b-d** are the statistical results of the linewidth after sintering, which may lead to the linewidth-power nonlinearity to be insignificant, since sintering shrinkage is nonlinear either **Fig. 1p**.

We had not previously focused on this issue and are very grateful to the reviewers for bringing it up.

We have added statements in the revised manuscript to briefly discuss this issue (**Page 7, highlighted in blue**), also as shown below:

.....“It should be noted that the scatter data in Fig. 2b-d tend to be more linear than those reported in the literature, which may be due to the introduction of BTPOS and the sintering shrinkage.”.....

Refs.

1. Liu, T., et al., Ultrahigh-printing-speed photoresists for additive manufacturing. *Nature Nanotechnology*, 2024. 19(1): 51-57.
2. Cao, C., et al., Click chemistry assisted organic-inorganic hybrid photoresist for ultra-fast two-photon lithography. *Additive Manufacturing*, 2022. 51: 102658.
3. Konstantinou, G., et al., Additive micro-manufacturing of crack-free PDCs by two-photon polymerization of a single, low-shrinkage preceramic resin. *Additive Manufacturing*, 2020. 35: 101343.
4. Zhou, X., Y. Hou, and J. Lin, A review on the processing accuracy of two-photon polymerization. *Aip Advances*, 2015. 5(3).

3. Figure S13a-c shows the deterioration of the edge roughness of the processed nanowires. The first reason explained by the author "may be due to the poor quality of the spot generated by the air objective lens." is not very reasonable. "Poor spot quality" generally refers to deformation of the beam shape or intensity distribution. One possible reason here is that the light intensity edge gradient of the focused spot of different NA objectives is different.

The intensity gradient of the air lens 0.95NA is smaller than that of the oil immersion objective, so the edge roughness of the exposure line worsens. Another reason is the standing wave effect proposed by the author. The standing wave will cause two phenomena. One is the decrease in adhesion of the polymer edge (the author has already mentioned), and the other is the difference in the development environment at the edge of the line. The standing wave effect causes Some small monomers are more fully developed than structures attached to the substrate.

Response:

We appreciate the comments. After consulting with experts in the field of optics, we recognize that you are absolutely correct. It should be the intensity gradient and not the spot quality that affects the edge roughness of the processed lines. The deformation of the beam shape or intensity distribution can only leads to a change in the 3D shape of the fabricated line.

Obviously, you know a lot about photolithography and the lithography process, thanks for popularizing your expertise to us. Since most of our authors are materials/chemistry majors, our understanding of optics is far from adequate, which resulted in the previous description not making sense. We deeply apologize that this may have misled the readers.

According to your suggestions, we have modified some statements in the revised manuscript to briefly discuss this issue (**Page 12, highlighted in blue**), also as shown below:

.....“Moreover, a deterioration in line edge roughness is also observed. One possible reason here is that the light intensity edge gradient of the focused spot of different NA objectives is different. The intensity gradient of the air objective (NA=0.95) is smaller than that of the oil objective (NA=1.45), so the edge roughness of the exposure line worsens. Another reason is the standing wave effect, which will not only cause the decrease in adhesion of the line edge, but also make a difference in the developing environment at the edge of the line. Some small monomers in photoresist are more fully developed than structures attached to the substrate.”.....

4. The contrast of the SEM photo in Figure 1(p) is so low that it is almost invisible. Please deal with it.

Response:

We appreciate the comments. We have deal with it in the revised manuscript. **Fig. 1p** has been replaced by a high-resolution one (as shown below), and the insert Figure has been re-arranged and moved to Supplementary Information (**Supplementary Fig. 9**, as shown below).

Fig. 1p The shrinkage rate of the ZnO patterns after pyrolysis, the inset shows the pattern used in the shrinkage test.

Supplementary Fig. 9 The shrinkage rate of the patterns with a CD below 1 μm , the insert shows a typical line before and after pyrolysis.

5. In addition to the transmittance curves of the three photoresists, can the absorbance curve of zinc-based, copper-based and zirconium-based photoresist be added?

Response:

We appreciate the comments. As requested, we have performed the additional experiments, and the absorbance curve of zinc-based, copper-based, zirconium-based photoresist and initiator DETC have been provided in the revised manuscript (**Supplementary Fig. 11**, as shown below).

It can be seen that zinc-based photoresist and copper-based photoresist have strong absorption in the deep UV region, while zirconium-based photoresist absorbs very little. Therefore, it is more likely for zinc-based photoresist and copper-based photoresist to absorb the femtosecond laser through higher order nonlinearities to achieve high-precision (CD), which is in agreement with our experimental results, i.e., 35 nm for Zn-based photoresist, 36 nm for Cu-based photoresist, and 58 nm for Zr-based photoresist (**Fig. 2b-f**).

We have added a short discussion in the revised manuscript to address this new finding (**Page 8-9, highlighted in blue**), also as shown below:

.....“Meanwhile, it also can be seen that Zn- / Cu-based precursor photoresist have strong absorption in the deep UV region (Supplementary Fig. 11), but not for Zr-based precursor photoresist, suggesting Zn- / Cu-based precursor photoresist are more likely to absorb the femtosecond laser through higher order nonlinearities (bigger N) to achieve high-precision.”.....

Supplementary Fig. 11 The absorption spectrum **a** zinc-based, **b** copper-based, **c** zirconium-based photoresist and **d** initiator DETC.

6. The author compared various MOS patterning preparation processes developed by other researchers in Table S3. It seems that the biggest contribution is the author's realization of MOS patterning below sub-100 nanometers. However, the authors did not demonstrate the improvement effect of this high-precision pattern on device preparation and performance. From the perspective of MOS device preparation alone, other methods (listed by the author in Table S3) can also be processed and do not require high-precision preparation methods. Therefore, it would be more breakthrough if the author could demonstrate the advantages of the proposed processing method in device preparation.

Response:

We appreciate the comments. This is a very meaningful suggestion. As requested, we have carried out the relevant experiments. In response to your concerns, we have made a comprehensive explanation below.

Supplementary Fig. 19 Optical microscope images (up: open-field mode, down: dark-field mode) of ZnO UV photodetector with different line-widths: **a** 0.5 μm , **b** 1 μm , **c** 2 μm . I–V characteristic curves of ZnO UV photodetector in the dark and under 365 nm UV illumination (0.1 mW cm^{-2}): **d** 0.5 μm , **e** 1 μm , **f** 2 μm .

Section A: High-precision pattern enables performance improvement.

As can be seen from **Supplementary Fig. 19**, while keeping the number of lines constant, as the line-width becomes smaller, the photo-to-dark current ratio (PDCR) increases dramatically. PDCR represents the signal-to-noise ratio, which is a key metric for sensors, thus illustrating that a high precision ROS is helpful in improving performance. Obviously, this performance improvement may also occur in other micro-devices.

Since it is difficult to process 35 nm ROS on a silicon substrate using an air objective with a low numerical aperture (NA0.95), and our oil objective (NA1.45) is not suitable for silicon substrate. So here we only provide linewidths from 0.5 μm to 2 μm , but its enough to illustrate the beneficial effect of high precision on performance. Also, after consulting with optical experts, once the line width is less than 1/4 of the wavelength (365 nm), it may be difficult for the ROS pattern to absorb photons efficiently. However, this does not mean that ultra-high precision ROS cannot be used in other miniature devices, especially microelectronic devices.

Although high precision is not necessary for some devices, if high precision is possible, it is obviously benefit to device miniaturization and integration, which is the trend of the future. This can not be completed by the processing method in **Supplementary Table 2**.

In addition to the effect of ROS precision on performance, we had tried to perform some ZnO UV photodetectors with various ROS pattern (**Fig. R2**, as shown below) before we submitted the original manuscript, so as to explore the effect of morphology on performance. This has not covered by the literature in **Supplementary Table 2**. Unfortunately, the performance didn't seem to make a difference, so that these results were not included in the manuscript. But, this may work in other devices.

Fig. R2 ZnO UV photodetectors with various ROS pattern.

Section B: Performance comparison with literature.

Obviously, the realization of ultra-high-precision MOS is just one of our important contributions, but not all. The ZnO UV photodetectors prepared by our strategy have superior performance. To demonstrate the advancement of our devices, we have listed the relevant literature [Ref.1-30] for comparison in the revised manuscript (**Supplementary Table 4**, as shown below). The superior performance of our devices can be clearly evidenced.

The possible reasons for the superior performance of the ZnO UV photodetectors prepared by our strategy are as follows:

The photoresist we use has a high metal content (**Supplementary Table 1**), and the resulting ROS microstructures have fewer defects (holes, discontinuities, low crystallinity, **Fig. 1-2**), which allows for higher performance. For example, we have prepared ZnO photodetectors with PDCR value up to 10500 (365 nm,) and 100000 (254 nm), respectively (**Supplementary Fig. S18** and **Fig. 4e**), which are higher than the majority of various ZnO photodetectors (including various structures) (**Supplementary Table 4**).

Meanwhile, homogeneous elemental doping (atomic level) can be easily achieved within our photoresist systems. For example, aluminum acrylate is mixed into the photoresist, and aluminum acrylate acts as a reactive monomer during the polymerization process and participates in the construction of a robust polymer network (**Supplementary Fig. S12a**), ensuring that the Al atoms are not lost / aggregate during the development / sintering process. This can facilitate the performance of the element-doped devices. Consequently, remarkable performance can be obtained by our strategy, e.g., the PDCR of Al-doped ZnO photodetector reaches to 16000 (10500 for pure ZnO photodetector) at 365 nm, which are higher than the element-doped ZnO photodetector (**Supplementary Table 4**).

Supplementary Table 4 The performance comparison of ZnO UV photodetectors.

Authors	Types of MOS	PDCR value	Journal, year	Ref.
Basavaraj G. Hunashimarad, et al.	Ca-doped ZnO film	3.17 at 365 nm	Optical Materials 2022	[1]
Sabina M. Hatch, et al.	ZnO-nanorods-CuSCN	4.5 at 375 nm	Advanced materials 2013	[2]
Shoou-Jinn Chang, et al.	Fe-doped ZnO	<10 at 375 nm	IEEE Photonics Technology Let. 2013	[3]
Qi Li, et al.	ZnO-CuO nanorod	10 at 325 nm	Advanced optical materials 2022	[4]
Chih-Hung Hsiao, et al.	Needle-like Ga-ZnO nanorods	11.07 at 360 nm	IEEE transactions on electron devices 2013	[5]
Jing Wang, et al.	ZnO nanowires	24.2 at 365 nm	J. Mater. Chem. C 2016	[6]
Chiung-Hsien Huang, et al.	Li-doped ZnO nanorods	34.87 at 380 nm	Microsystem Technologies 2022	[7]
Sunghoon Park, et al.	ZnO nanowires	49 at 365 nm	Journal of Alloys and Compounds 2016	[8]
Min Chen, et al.	ZnO hollow-sphere nanofilm	53 at 350 nm	Small 2011	[9]
Huihui Yu, et al.	Atomic-thin ZnO Sheet	69.6 at 365 nm 120.1 at 254 nm	Small 2020	[10]
Cheng-Liang, Hsu et al.	Vertical ZnO nanowires	67.5 at 254 nm	Chemical Physics Letters 2005	[11]
Soo Hyun Lee, et al.	ZnO nanorods	1720 at 380 nm	Nanoscale Research Letters 2016	[12]
Shaivalini Singh, et al.	Al-doped ZnO	3327.94 at 372 nm	MicrosystemTechnologies 2016	[13]
Nishant Kumar, et al.	Mg-doped ZnO films	71.68 at 365 nm	Journal of Alloys and Compounds 2018	[14]
Nishant Kumar, et al.	Cd-doped ZnO	93.78 at 386 nm	Journal of Alloys and Compounds 2017	[15]
S.J. Young, et al.	ZnO film	290 at 370 nm	Sensors and Actuators A: Physical 2007	[16]
Yen-Lin Chu, et al.	Ni-doped ZnO	393.04 at 380nm	J. Electrochem. Soc. 2020	[17]
Fatemeh Abbasi, et al.	Ni-doped ZnO film	416.14 at 350 nm	Optics Communications 2021	[18]
Ramazanali Dalvand, et al.	ZnO nanoneedles	600 at 325nm	Journal of Materials Science: Materials in Electronics 2018	[19]
Liu kw, et al.	Mg-doped ZnO film	Approx. 1000 at 368 nm	Sensors 2010	[20]
Akshta Rajan, et al.	ZnO thin film	1000 at 365 nm	MRS Online Proceedings Library 2013	[21]
Hsiang-Chun Wang, et al.	Ag nanoparticles modified ZnO	1000 at 365 nm	Nanoscale Research Letters 2020	[22]
Zeping Li, et al.	ZnO quantum dot	1767.8 at 350 nm	Applied Surface Science 2022	[23]
Jingwei Liu, et al.	ZnO nanowire	3000 at 365	Adv. Mater. Technol. 2022	[24]
James Taban Abdalla, et al.	ZnO nanorod	4000 at 365 nm	Journal of Electronic Materials 2020	[25]
Fa Cao, et al.	ZnO-CuI heterostructure	4250 at 365nm	Journal of Alloys and Compounds 2021	[26]
Dawit Gedamuv, et al.	ZnO nanotetrapod networks	4500 at 365 nm	Advanced Materials 2014	[27]

Omar F. Farhat, et al.	ZnO nanoaggregates	8345 at 365 nm	Sensors and Actuators A: Physical 2021	[28]
Amit Kumar Rana, et al.	Co3O4-ZnO film	45700 at 365 nm	Materials Science in Semiconductor Processing 2020	[29]
Jin Hyung Jun, et al.	ZnO nanoparticles	10 ⁶ at 325 nm	Ceramics International 2009	[30]
	ZnO wire	10500 at 365 nm		This work
	Al-doped ZnO wire	16000 at 365 nm		This work
	ZnO wire	100000 at 254 nm		This work

Section C: Some necessary explanation

We want to emphasize that, our original purpose is to provide the readers or researchers a strategy that enables ultra-high-precision MOS fabrication, homogeneous elemental doping, heterogeneous structure preparation, and large area array preparation, giving them a wider range of choices in their devices.

In fact, we would prefer to use our strategy to process a real integrated circuit chip unit, such as transistors, logic gates, etc., but this requires very demanding experimental conditions that we are not able to realize for the time being, so we only show ZnO sensors to illustrate the practicality of our strategy.

Since our original focus is not on performance enhancement of fabricated devices, so we have overlooked the potential interest of some readers.

Thank you very much for your valuable suggestions and reminders. Supplementing the experiments and revising the manuscript as you requested not only further emphasizes the advancement of the manuscript, but also arouses the interest and attention of a wider audience.

We have added **Supplementary Fig. 19** and **Supplementary Table 4** into the “**Supplementary Information**” file, and have added some statement in the revised manuscript (**Page 13, highlighted in blue**), also as shown below:

.....“Notably, the superior performance of our devices can be clearly demonstrated by comparison with similar literature (Supplementary Table 4). Meanwhile, the PDCR of ZnO UV photodetector might be further improved by our strategy to reduce the line-width of ZnO (Supplementary Fig. 19).”.....

Refs.

-
- [1]. Hunashimarad, B.G., et al., ZnO:Ca MSM ultraviolet photodetectors. *Optical Materials*, 2022. 124: p. 111960.
- [2]. Hatch, S.M., J. Briscoe, and S. Dunn, A Self-Powered ZnO-Nanorod/CuSCN UV Photodetector Exhibiting Rapid Response. *Advanced Materials*, 2013. 25(6): p. 867-871.
- [3]. Chang, S.J., et al., Noise Properties of Fe-ZnO Nanorod Ultraviolet Photodetectors. *IEEE Photonics Technology Letters*, 2013. 25(21): p. 2089-2092.
- [4]. Li, Q., et al., Enhanced Performance of a Self-Powered ZnO Photodetector by Coupling LSPR-Inspired Pyro-Phototronic Effect and Piezo-Phototronic Effect. *Advanced Optical Materials*, 2022. 10(7): p. 2102468.
- [5]. Hsiao, C.H., et al., Field-Emission and Photoelectrical Characteristics of Ga-ZnO Nanorods Photodetector. *IEEE Transactions on Electron Devices*, 2013. 60(6): p. 1905-1910.
- [6]. Wang, J., et al., Ligand-directed rapid formation of ultralong ZnO nanowires by oriented attachment for UV photodetectors. *Journal of Materials Chemistry C*, 2016. 4(24): p. 5755-5765.
- [7]. Huang, C.-H., et al., Fabrication and characterization of homostructured photodiodes with Li-doped ZnO nanorods. *Microsystem Technologies*, 2022. 28(1): p. 369-375.
- [8]. Park, S., et al., ZnO-core/ZnSe-shell nanowire UV photodetector. *Journal of Alloys and Compounds*, 2016. 658: p. 459-464.
- [9]. Chen, M., et al., ZnO Hollow-Sphere Nanofilm-Based High-Performance and Low-Cost Photodetector. *Small*, 2011. 7(17): p. 2449-2453.
- [10]. Yu, H., et al., Atomic-Thin ZnO Sheet for Visible-Blind Ultraviolet Photodetection. *Small*, 2020. 16(47): p. 2005520.
- [11]. Hsu, C.-L., et al., Ultraviolet photodetectors with low temperature synthesized vertical ZnO nanowires. *Chemical Physics Letters*, 2005. 416(1): p. 75-78.
- [12]. Lee, S.H., S.H. Kim, and J.S. Yu, Metal-Semiconductor-Metal Near-Ultraviolet (~380 nm) Photodetectors by Selective Area Growth of ZnO Nanorods and SiO₂ Passivation. *Nanoscale Research Letters*, 2016. 11(1): p. 333.
- [13]. Singh, S., Al doped ZnO based MISIM ultraviolet photodetectors. *Microsystem Technologies*, 2017. 23(4): p. 999-1003.
- [14]. Kumar, N. and A. Srivastava, Green photoluminescence and photoconductivity from screen-printed Mg doped ZnO films. *Journal of Alloys and Compounds*, 2018. 735: p. 312-318.
- [15]. Kumar, N. and A. Srivastava, Faster photoresponse, enhanced photosensitivity and photoluminescence in nanocrystalline ZnO films suitably doped by Cd. *Journal of Alloys and Compounds*, 2017. 706: p. 438-446.
- [16]. Young, S.J., et al., ZnO-based MIS photodetectors. *Sensors and Actuators A: Physical*, 2007. 135(2): p. 529-533.
- [17]. Chu, Y.-L., et al., Fabrication and Characterization of Ni-Doped ZnO Nanorod Arrays for UV Photodetector Application. *Journal of The*

-
- Electrochemical Society, 2020. 167(6): p. 067506.
- [18]. Abbasi, F., F. Zahedi, and M.h. Yousefi, Fabricating and investigating high photoresponse UV photodetector based on Ni-doped ZnO nanostructures. *Optics Communications*, 2021. 482: p. 126565.
- [19]. Dalvand, R., S. Mahmud, and R. Shabannia, Fabrication of UV photodetector using needle-shaped ZnO nanostructure arrays prepared on porous silicon substrate by a facile low-temperature method. *Journal of Materials Science: Materials in Electronics*, 2018. 29(6): p. 4999-5008.
- [20]. Liu, K., M. Sakurai, and M. Aono ZnO-Based Ultraviolet Photodetectors. *Sensors*, 2010. 10, 8604-8634 DOI: 10.3390/s100908604.
- [21]. Rajan, A., et al., Plasmonic Enhancement of Optical Absorption of UV Radiation in ZnO Thin Film Based Ultraviolet Photodetectors. *MRS Online Proceedings Library*, 2013. 1509(1): p. 1212.
- [22]. Wang, H.-C., et al., ZnO UV Photodetectors Modified by Ag Nanoparticles Using All-Inkjet-Printing. *Nanoscale Research Letters*, 2020. 15(1): p. 176.
- [23]. Li, Z., et al., High performance ZnO quantum dot (QD)/ magnetron sputtered ZnO homojunction ultraviolet photodetectors. *Applied Surface Science*, 2022. 582: p. 152352.
- [24]. Liu, J., et al., 3D Printing Nano-Architected Semiconductors Based on Versatile and Customizable Metal-Bound Composite Photoresins. *Advanced Materials Technologies*, 2022. 7(6): p. 2101230.
- [25]. Abdalla, J.T., et al., Enhanced Ag@SnO₂ Plasmonic Nanoparticles for Boosting Photoluminescence and Photocurrent Response of ZnO Nanorod UV Photodetectors. *Journal of Electronic Materials*, 2020. 49(9): p. 5657-5665.
- [26]. Cao, F., et al., High-performance, self-powered UV photodetector based on Au nanoparticles decorated ZnO/CuI heterostructure. *Journal of Alloys and Compounds*, 2021. 859: p. 158383.
- [27]. Gedamu, D., et al., Rapid Fabrication Technique for Interpenetrated ZnO Nanotetrapod Networks for Fast UV Sensors. *Advanced Materials*, 2014. 26(10): p. 1541-1550.
- [28]. Farhat, O.F., et al., Tape-based novel ZnO nanoaggregates photodetector. *Sensors and Actuators A: Physical*, 2021. 332: p. 113210.
- [29]. Rana, A.K., et al., Transparent Co₃O₄/ZnO photovoltaic broadband photodetector. *Materials Science in Semiconductor Processing*, 2020. 117: p. 105192.
- [30]. Jun, J.H., et al., Ultraviolet photodetectors based on ZnO nanoparticles. *Ceramics International*, 2009. 35(7): p. 2797-2801.

Reviewer #2 (Remarks to the Author):

The submitted manuscript “Ultra-high Precision Nano Additive Manufacturing of Metal Oxides Semiconductor” by Chun Cao et al. describes a lithography method using multiphoton polymerization followed by a pyrolysis step to print narrow lines of metal oxide materials. Notably, the authors use a spincoating photoresist strategy rather than a drop casting method in an attempt to reduce the mobility of radicals formed during printing. After exposure with a typical multiphoton lithography printing scheme, the printed structures are pyrolyzed under high temperature to remove organic components leaving a metal oxide semiconducting material. The pyrolysis step results in shrinkage on the order of 20% leading to linewidths of oxide material of range 35 – 72 nm. The authors also present one doped metal oxide as well as use a printed oxide for UV photodetection. While the authors present some stacked layers of materials, they do not show any truly 3D structures typically associated with the multiphoton lithography process. It is not clear from what is presented what is limiting this process from showing a truly 3D structure. While the authors show an impressive linewidth capability (critical dimension, CD) of laser printed oxide material they do not explicitly demonstrate an example of the advantage of this fine feature size. All examples of devices are printed with dimensions that can be achieved with previously demonstrated methods in the literature (see references from the Figure 2i). Overall, the paper is organized well and easy to follow. The results described here may have some interest to the community. I have outlined below points and concerns that the authors should address if this article is to be considered for publication.

Response:

We appreciate your positive comments and constructive suggestions, which certainly help us to improve the quality of our manuscript. We note that you have two main concerns: 3D printing capability and the advantages of high-precision lines for device preparation.

In response to the first concern, we have supplemented the experiment, and provided several truly 3D structures (**Supplementary Fig. 15**) in the revised manuscript, demonstrating the 3D fabricating ability of our strategy. Since our original attention is on the ability of high-precision ROS fabrication,

and 3D fabrication was overlooked in the previous manuscript. For 3D structures fabrication, the photoresist should be prepared with a high content (see **Methods section**) and drop-casted onto substrate to achieve enough film thickness. We have made a brief discussion about this issue in the revised manuscript (**Page 12, highlighted in blue**).

Supplementary Fig. 15 3D micro-structures fabricated by our strategy using zirconium-based photoresist. **a** Nut array (the thread on the side can be clearly seen.) and **b** diamond cubic lattice structure.

In terms of the advantages of high-precision lines for device preparation, we have explored the effect of line width on performance (**Supplementary Fig. 19**), and made a comprehensive explanation below.

Section A: High-precision pattern enables performance improvement.

As can be seen from **Supplementary Fig. S19**, while keeping the number of lines constant, as the line-width becomes smaller, the photo-to-dark current ratio (PDCR) increases dramatically. PDCR represents the signal-to-noise ratio, which is a key metric for sensors, thus illustrating that a high precision ROS is helpful in improving performance. Obviously, this performance improvement may also occur in other micro-devices.

Since it is difficult to process 35 nm ROS on a silicon substrate using an air objective with a low numerical aperture (NA0.95), as mentioned by reviewer #1, and our oil objective (NA1.45) is not suitable for silicon substrate. So here we only provide linewidths from 0.5 μm to 2 μm, but its enough to illustrate the beneficial effect of high precision on performance. Also, after consulting with optical experts, once the line width is less than 1/4 of the wavelength (365 nm), it may be difficult for the ROS pattern to absorb photons efficiently. However,

this does not mean that ultra-high precision ROS cannot be used in other miniature devices, especially microelectronic devices.

Although high precision is not necessary for some devices, if high precision is possible, it is obviously benefit to device miniaturization and integration, which is the trend of the future. This can not be completed by the processing method in Table S3.

Supplementary Fig. 19 Optical microscope images (up: open-field mode, down: dark-field mode) of ZnO UV photodetector with different line-widths: **a** 0.5 μm, **b** 1 μm, **c** 2 μm. I–V characteristic curves of ZnO UV photodetector in the dark and under 365 nm UV illumination (0.1 mW cm⁻²): **d** 0.5 μm, **e** 1 μm, **f** 2 μm.

Section B: Performance comparison with literature.

Obviously, the realization of ultra-high-precision MOS is just one of our important contributions, but not all. The ZnO UV photodetectors prepared by our strategy have superior performance. To demonstrate the advancement of our devices, we have listed the relevant literature for comparison in the revised manuscript (**Supplementary Table 4**, also as shown below). The superior performance of our devices can be clearly evidenced.

The possible reasons for the superior performance of the ZnO UV photodetectors prepared by our strategy are as follows:

The photoresist we use has a high metal content (**Supplementary Table 1**), and the resulting ROS microstructures have fewer defects (holes, discontinuities, low crystallinity, **Fig. 1-2**), which allows for higher performance. For example, we have prepared ZnO photodetectors with PDCR value up to 10500 (365 nm,) and 100000 (254 nm), respectively (**Supplementary Fig. S18a, Fig. 4e and Supplementary Table 4**), which are higher than the majority of pure ZnO photodetectors (including various structures) (**Supplementary Table 4**).

Meanwhile, homogeneous elemental doping (atomic level) can be easily achieved within our photoresist systems. For example, aluminum acrylate is mixed into the photoresist, and aluminum acrylate acts as a reactive monomer during the polymerization process and participates in the construction of a robust polymer network (**Supplementary Fig. 12a**), ensuring that the Al atoms are not lost / aggregate during the development / sintering process. This can facilitate the performance of the element-doped devices. Consequently, remarkable performance can be obtained by our strategy, e.g., the PDCR of Al-doped ZnO photodetector reaches to 16000 (10500 for pure ZnO photodetector) at 365 nm, which are higher than the element-doped ZnO photodetector (**Supplementary Table 4**).

The ZnO photodetectors prepared by the existing techniques in **Supplementary Table 2** do not have the above mentioned advantages, and hence their performance is hardly superior to ours.

Supplementary Table 4 The performance comparison of ZnO UV photodetectors.

Authors	Types of MOS	PDCR value	Journal, year	Ref.
Basavaraj G. Hunashimarad, et al.	Ca-doped ZnO film	3.17 at 365 nm	Optical Materials 2022	[1]
Sabina M. Hatch, et al.	ZnO-nanorods-CuSCN	4.5 at 375 nm	Advanced materials 2013	[2]
Shouu-Jinn Chang, et al.	Fe-doped ZnO	<10 at 375 nm	IEEE Photonics Technology Let. 2013	[3]
Qi Li, et al.	ZnO-CuO nanorod	10 at 325 nm	Advanced optical materials 2022	[4]
Chih-Hung Hsiao, et al.	Needle-like Ga-ZnO nanorods	11.07 at 360 nm	IEEE transactions on electron devices 2013	[5]
Jing Wang, et al.	ZnO nanowires	24.2 at 365 nm	J. Mater. Chem. C 2016	[6]

Chiung-Hsien Huang, et al.	Li-doped ZnO nanorods	34.87 at 380 nm	Microsystem Technologies 2022	[7]
Sunghoon Park, et al.	ZnO nanowires	49 at 365 nm	Journal of Alloys and Compounds 2016	[8]
Min Chen, et al.	ZnO hollow-sphere nanofilm	53 at 350 nm	Small 2011	[9]
Huihui Yu, et al.	Atomic-thin ZnO Sheet	69.6 at 365 nm 120.1 at 254 nm	Small 2020	[10]
Cheng-Liang, Hsu et al.	Vertical ZnO nanowires	67.5 at 254 nm	Chemical Physics Letters 2005	[11]
Soo Hyun Lee, et al.	ZnO nanorods	1720 at 380 nm	Nanoscale Research Letters 2016	[12]
Shaivalini Singh, et al.	Al-doped ZnO	3327.94 at 372 nm	Microsystem Technologies 2016	[13]
Nishant Kumar, et al.	Mg-doped ZnO films	71.68 at 365 nm	Journal of Alloys and Compounds 2018	[14]
Nishant Kumar, et al.	Cd-doped ZnO	93.78 at 386 nm	Journal of Alloys and Compounds 2017	[15]
S.J. Young, et al.	ZnO film	290 at 370 nm	Sensors and Actuators A: Physical 2007	[16]
Yen-Lin Chu, et al.	Ni-doped ZnO	393.04 at 380nm	J. Electrochem. Soc. 2020	[17]
Fatemeh Abbasi, et al.	Ni-doped ZnO film	416.14 at 350 nm	Optics Communications 2021	[18]
Ramazanali Dalvand, et al.	ZnO nanoneedles	600 at 325nm	Journal of Materials Science: Materials in Electronics 2018	[19]
Liu kw, et al.	Mg-doped ZnO film	Approx. 1000 at 368 nm	Sensors 2010	[20]
Akshta Rajan, et al.	ZnO thin film	1000 at 365 nm	MRS Online Proceedings Library 2013	[21]
Hsiang-Chun Wang, et al.	Ag nanoparticles modified ZnO	1000 at 365 nm	Nanoscale Research Letters 2020	[22]
Zeping Li, et al.	ZnO quantum dot	1767.8 at 350 nm	Applied Surface Science 2022	[23]
Jingwei Liu, et al.	ZnO nanowire	3000 at 365	Adv. Mater. Technol. 2022	[24]
James Taban Abdalla, et al.	ZnO nanorod	4000 at 365 nm	Journal of Electronic Materials 2020	[25]
Fa Cao, et al.	ZnO-CuI heterostructure	4250 at 365nm	Journal of Alloys and Compounds 2021	[26]
Dawit Gedamuv, et al.	ZnO nanotetrapod networks	4500 at 365 nm	Advanced Materials 2014	[27]
Omar F. Farhat, et al.	ZnO nanoaggregates	8345 at 365 nm	Sensors and Actuators A: Physical 2021	[28]
Amit Kumar Rana, et al.	Co ₃ O ₄ -ZnO film	45700 at 365 nm	Materials Science in Semiconductor Processing 2020	[29]
Jin Hyung Jun, et al.	ZnO nanoparticles	10 ⁶ at 325 nm	Ceramics International 2009	[30]
	ZnO wire	10500 at 365 nm	This work	
	Al-doped ZnO wire	16000 at 365 nm	This work	
	ZnO wire	100000 at 254 nm	This work	

Section C: Some necessary explanation

We want to emphasize that, our original purpose is to provide the readers or researchers a strategy that enables ultra-high-precision MOS fabrication, homogeneous elemental doping, heterogeneous structure preparation, and large area array preparation (the techniques described in **Supplementary Table 2** are difficult to realize or are not mentioned), giving readers a wider range of choices in their devices.

In fact, we would prefer to use our strategy to process a real integrated circuit chip unit, such as transistors, logic gates, etc., but this requires very demanding experimental conditions that we are not able to realize for the time being, so we only show ZnO sensors to illustrate the practicality of our strategy.

Thank you very much for your valuable suggestions and reminders. Supplementing the experiments and revising the manuscript as you requested not only further emphasizes the advancement of the manuscript, but also arouses the interest and attention of a wider audience.

We have added **Supplementary Fig. S19** and **Supplementary Table 4** into the “**Supplementary Information**” file, and have added some statement in the revised manuscript (**Page 13, highlighted in blue**).

1. The abbreviations DUV and EUV are used before defining them.

Response:

We appreciate the comments. We have defined them when they first appeared in the revised manuscript (**Page 1, highlighted in blue**).

2. The authors seem to use the term critical dimension (CD) for both the width of a polymerized line before pyrolysis and after pyrolysis. This becomes confusing, especially when trying to understand Figure 1p. The authors should better differentiate between the measurements taken during these two different steps of their whole process.

Response:

We appreciate the comments. We apologize that we have not distinguished this matter well before, and in so doing have caused distress to our readers.

CD generally refers to the smallest size (line width or diameter of a dot) of a structure that can be machined, and is sometimes referred to as the feature size or half-pitch. Therefore, all the “CD” for “lines” and “rectangular patterns” that before pyrolysis have been replaced by “line-width” or “side length” or “size” or “precision” in the revised manuscript (In particular, page 6-7, highlighted in blue). And the remaining “CD” are only used to characterize the samples after sintering.

3.The insets of Figure 1p are very hard to see. These may be better as their own figure, maybe in the supplementary info.

Response:

We appreciate the comments. We have deal with it in the revised manuscript. **Fig. 1p** has been replaced by a high-resolution one (as shown below), and the insert Figure has been re-arranged and moved to Supplementary Information (**Supplementary Fig. S9**, as shown below).

Fig. 1p The shrinkage rate of the ZnO patterns after pyrolysis, the inset shows the pattern used in the shrinkage test.

Supplementary Fig. S9 The shrinkage rate of the patterns with a CD below 1 μm, the insert shows a typical line before and after pyrolysis.

4. It seems odd to present the atomic ratios of only select atoms such that they add up to 100%. Is this typically how it is done? It seems to me that you can still show the ratio of the atoms you are interested in without excluding other atoms from the measurement.

Response:

We appreciate the comments. As requested, we have provided the real atomic ratios of Zn and O in **Fig. 1i**, that are 31.11% and 31.13%, respectively, demonstrating the fabrication of ZnO. Besides, the atomic ratios of Si, Pt, and C are 26.09%, 5.37%, and 2.30%, respectively.

Trace C (atomic ratio: 2.30%) is mainly from combustion residue and SEM sediment contamination. Si derives from the silicon substrate, and Pt comes from the sputtered conductive layer on the sample surface. We have added the atomic ratios of Zn, O, and C in the revised manuscript (**page 5, highlighted in blue**).

5. In figure 1h, it seems to me that there is still a little bit of carbon signal in the shape of the grid of printed material. Can the authors better specify the purity of the after pyrolysis sample?

Response:

We appreciate the comments. As mentioned above, the atomic ratio of residual C elements is 2.30%, part of which comes from unavoidable SEM

sediment contamination. We did not find any diffraction peaks, diffraction rings and lattices of C in XRD, SAED and high-resolution TEM, which suggests a very limited amount of C in the fabricated ZnO patterns.

Of course, if the readers want to get less C in ROS, they can optimize the sintering process to achieve. For example, increase the sintering time, reduce the rate of temperature increase, increase the oxygen content in the sintering atmosphere, stay at a certain temperature for some time, etc..

6. The authors state " When the CD is small enough, it can be regarded as a three-dimensional shrinkage, which can be up to 20%". Do the authors do any measurements of the vertical/height shrinkage? I don't see any in the manuscript. Then this statement is confusing.

Response:

We appreciate the comments. We apologize for giving this confusing statement in the previous manuscript. Since we find that, when the sizes below 1 μm , the horizontal shrinkage no longer seems to be size-dependent and varies in the range of 12-20 % (**Supplementary Fig. 9**).

What we originally intended to express was that when the CD is small enough (less than 1 μm), the fabricated structure might be of the same order of magnitude in terms of its dimensions in the horizontal and vertical directions, and the structure is more like a "mini 3D structure". At this point, the substrate's restriction on the structure may be weakened, resulting in a non-size-dependent shrinkage.

We recognize that this statement is only our speculation and lacks experimental support, so we have modified it in the revised manuscript (**page 6, highlighted in blue**), as shown below.

".....At sizes below 1 μm , the shrinkage no longer seems to be size-dependent and varies in the range of 12-20%. (Supplementary Fig. 9)....."

7. I did not find the height of the spin-coated material in the manuscript. What is the thickness?

Response:

We appreciate the comments. We have provided the thickness of the spin-coated photoresist in the revised manuscript (**Experimental Section, highlighted in blue**).

It is worth noting that the film thickness is highly dependent on the spin-coating process and photoresist formulation (content of solutes, type of solvent, etc.). In our conditions, the film thickness is approx. 30 nm (tested by SEM observation from the cross section, as shown below) for the three kind of photoresists.

Fig. R3 The thickness of the spin-coated photoresists for high-precision ROS fabrication (tested by SEM observation from the cross section).

8 Did the authors prebake the spin-coated substrate before exposure?

Response:

We appreciate the comments. For 2D ROS fabrication, We did not prebake the spin-coated samples (Zn- / Cu / Zr-based precursor photoresist), since the spin-coated photoresist is very thin (approx. 30 nm), the vast majority of the solvent has evaporated during the spin-coating process and the residual solvent does not affect the exposure. For micro 3D MOS fabrication, the as-prepared photoresist was drop-casted onto a glass substrate and then dried at 50 °C for 2 h. We have mentioned these in the revised manuscript (**Methods section**).

For the commercial photoresist AZ1050 used in our device preparation, we had performed pre-bake and post-bake according to its instructions, and the relevant parameters have been added to the revised manuscript (**page 16-17, highlighted in blue**).

9. How are the samples developed before pyrolysis?

Response:

We appreciate the comments. Development is essential before pyrolysis, otherwise unexposed photoresist will also remain, making it impossible to distinguish the exposure pattern. We apologize for the previous omission, and we have added the development parameters in the revised manuscript, that is “*After MPL, metal-based precursor photoresists were developed with PGMEA for 30 s and then rinsed with IPA for 60 s.*” (page 17, highlighted in blue).

10. The authors in multiple places explain the effect of BTPOS as a scavenger of radicals in their system. Have the authors actually proved that BTPOS behaves this way in their system and has an effect on the CD?

Response:

We appreciate the comments. BTPOS has been introduced into photoresists as a scavenger (quencher or free radical trapping agent) of radicals for many times. And we had cited the relevant literature [Ref.1-3] in the previous manuscript when first mentioned in previous manuscript (page 3, highlighted in blue).

Furthermore, in our recent publication [Ref.4], we have investigated in detail the mechanism (three quenching paths, Fig. R4) of free radical quenching by nitrogen-oxygen-like molecules (including, but not limited to, **BBTPOS**. They have similar functional group, i.e., the N-O* group. (Fig. R5)).

It is worth noting that the same photosensitizer (DETC) is used in Ref.4 and the present work, and the active resins are both acrylic systems (Fig. R5), with free radicals initiating the polymerization of carbon-carbon double bonds. Therefore, the role played by BTPOS is obvious, and there is no need for us to do further exploration in the present work.

In addition, different scavengers (quenchers or free radical trapping agents) have different compatibility with resins, and we chose BTPOS over other quenchers because of its good compatibility with metal-based precursor photoresists, which will not precipitate out during spin-coating.

Fig. R4 Quenching mechanism of nitrogen-oxygen-like molecules in matter confined MPL. [Ref.4]

Fig. R5 The chemical structures of different quenchers (Q-4 is BTPOS) and the used resins and initiator. [Ref.4]

Refs.

1. Hahn, V, et al., Two-step absorption instead of two-photon absorption in 3D nanoprining. *Nature Photonics* 2021, 15, 932-938.
2. Mayer, F, et al., 3D Two-Photon Microprinting of Nanoporous Architectures. *Adv Mater* 2020, 32, e2002044.
3. Takada, K, et al., Improved spatial resolution and surface roughness in photopolymerization-based laser nanowriting. *Applied Physics Letters* 2005, 86, 1864249.
4. Guan, L., et al., Light and matter co-confined multi-photon lithography. *Nature Communications*, 2024. 15(1): 2387.

11. In the main text, the authors state that the photoinitiator concentration is the same for all photoresists. However, this does not seem to be the case when looking at the amounts described in the Methods section of the paper. Please address this. If the concentrations are not the same, it makes much of the discussion comparing CD of the different materials irrelevant.

Response:

We appreciate the comments. We are sure that the photoinitiator concentration is the same for all photoresists (same ratio of initiator / metal-organic compounds / solvent), and there is a mistake in the Methods section. We have fixed the error in the revised manuscript.

The details are as follows:

“Preparation of zinc-based precursor photoresist : “Mixed zinc methacrylate (150 mg) and DETC (3 mg) into PGMEA (3.0 g)”

Preparation of copper-based precursor photoresist: “Mixed copper (II) acrylate (250 mg) and DETC (5 mg) into PGMEA (5.0 g)

Preparation of zirconium-based precursor photoresist: “Mixed zirconium bromonorborelactone carboxylate triacrylate (250 mg) and DETC (5 mg) into PGMEA (5.0 g)”

Preparation of aluminum-doped zinc-based precursor photoresist: “Zinc methacrylate (148.5 mg), aluminum acrylate (1.5 mg) and DETC (3 mg) into PGMEA (3.0 g)”

It can be seen that the mass ratio of metal-organic compounds / initiator / solvent is constant at 50 : 1 : 1.

We apologize for the previous oversight.

12. Figures 2g and 2h are introduced out of order in the text. Consider rearranging the panels.

Response:

We appreciate the comments. We have re-arranged the panels of **Fig. 2** in the revised manuscript.

13. The authors discuss DFT calculations of the metal complexes (Table S1) and link the excitation energies to the observed higher N values in their resists. Is it then supposed to be understood that the absorption of the metal complexes is significant compared to the photoinitiator ? Why do the authors bother using a photoinitiator if the resist absorbs without it ? It seems to me that an argument about diffusion would make more sense here, especially considering the papers that the authors cite for this argument (47, 65).

Response:

We appreciate the valuable comments. To address your concerns, we have made a comprehensive explanation below.

Section A: The contribution of the resin and initiator to N values.

We believe that the N values obtained from our experimental tests should be the result of the joint contributions of photoinitiator and metal complexes. In fact, this had been stated directly or indirectly in both of the literatures (47, 65) you mentioned.

For example, in “Fischer, J.; Mueller, J. B.; Kaschke, J.; Wolf, T. J.; Unterreiner, A. N.; Wegener, M. Three-dimensional multi-photon direct laser writing with variable repetition rate. *Opt Express* 2013, 21, 26244-26260.”, it had reported that “*Although we do not claim to fully understand or model the photoresist system, this $N = 7$ process appears consistent with a multi-photon photoionization of the monomer molecules creating the initiating species.*”, as shown in **Fig. R6**.

Likewise, in “Yang, L.; Münchinger, A.; Kadic, M.; Hahn, V.; Mayer, F.; Blasco, E.; Barner-Kowollik, C.; Wegener, M. On the Schwarzschild Effect in 3D Two-Photon Laser Lithography. *Advanced Optical Materials* 2019, 7, 1901040.”, it had claimed that “*For the extreme case of zero photoinitiator concentration, i.e., for using pure PETA, $N = 4.2 \approx 4$ provides the best fit to the data. This behavior indicates that the monomer itself can be excited by four-photon absorption. Therefore, the overall behavior is a combination of two-photon and four-photon absorption, effectively leading to intermediate and noninteger values of N from the fits.*”, as shown in **Fig. R7**.

[Figure redacted]

Fig. R6 The data and text from “Opt Express 2013, 21, 26244-26260.”.

[Figure redacted]

Fig. R7 The text from “Advanced Optical Materials 2019, 7, 1901040.”.

Therefore, based on our experimental and DFT results, we believe that the resins (metal-organic compounds) do contribute to N value, and further influence the lithography performance. However, we must recognize that we cannot quantify the contribution of resins and initiators to N, nor can we declare who contributes more.

The absorption spectrum (**Supplementary Fig. 11**, as shown below) of the photoresists and initiator might provide some clues. It can be seen that zinc-based photoresist and copper-based photoresist have strong absorption in the deep UV region, while zirconium-based photoresist absorbs very little. Therefore, it is more likely for zinc-based photoresist and copper-based photoresist to absorb the femtosecond laser through higher order nonlinearities (bigger N) to achieve high-precision, which is in agreement with our

experimental results, i.e., 35 nm for Zn-based photoresist, 36 nm for Cu-based photoresist, and 58 nm for Zr-based photoresist (**Fig. 2b-f**).

Supplementary Fig. 11 The absorption spectrum **a** zinc-based, **b** copper-based, **c** zirconium-based photoresist and **d** initiator DETC.

However, absorption does not mean that reactive radicals can be generated to initiate polymerization. For most pure resins (including PETA that was used in the above two literatures), which have a weak charge transfer capacity to produce enough free radicals, tend to explode / damage at higher laser power rather than photo-polymerization.

For our photoresists, without the initiator DETC, they also tend to explode or damage, indicating that the metal complexes are also not able to produce enough reactive radicals. Thus, the initiator is necessary for our photoresists.

Notably, there may be an energy transfer between metal complexes and DETC, as we found that our photoresists lost STED (stimulated emission dissipation) capability. By absorbing a femtosecond laser, the metal complexes in the excited state might be able to transfer energy to the ground state DETC

(S₀), and then the excited DETC (S_n) undergoes intersystem crossing to produce free radicals. This process may lead to STED failure, and this is why we didn't use STED lithography to achieve a smaller CD. We are looking into this and hope to figure out why STED fails.

We have added a short discussion in the revised manuscript to discussion the absorption spectrum (**Page 8, highlighted in blue**).

Section B: The role of diffusion in MPL.

Of course, both of the above two papers had also discussed or stated the effect of diffusion on precision. For example, in "Opt Express 2013, 21, 26244-26260.", it had reported that "*This may be an indication that the resolution of this photoresist is actually limited by effects like diffusion and not by optics.*", as shown in **Fig. R8**.

[Figure redacted]

Fig. R8 The text from "Opt Express 2013, 21, 26244-26260."

Likewise, in "Advanced Optical Materials 2019, 7, 1901040.", it had claimed that "*We have explained this behavior in terms of the diffusion of oxygen, which serves as a quencher, and the diffusion of photoinitiator molecules.*" and "*where the polymerization threshold is lowered due to oxygen depletion via diffusion into previously exposed regions. If the photoinitiator concentration has been (unintentionally) depleted locally by long low-power illumination, times on the order of seconds have to be planned for before diffusion recovers photoinitiator concentrations that allow for normal writing.*", as shown in **Fig. R9**.

[Figure redacted]

Fig. R9 The text from “Advanced Optical Materials 2019, 7, 1901040.”.

Overall, the effect of diffusion on precision has two main aspects, one is the diffusion of free radicals produced by the initiator from the exposure area outward, and the other is the diffusion of quenchers (including BTPOS and oxygen) from the outside into the exposure area. Obviously, the former will cause the line width to become larger and the latter compresses it.

Indeed, we have already emphasized the role of diffusion in CD in the manuscript, that is “*Although MPL can exceed the diffraction limit and achieve an feature size of nearly 100 nm with the aid of nonlinear absorption, the migration of free radicals hinders the further improvement of precision. In addition to utilizing solid-state precursor photoresists to reduce free radical migration, we have introduced free radical trapping agent (Fig. 1a, BTPOS), whereby free radicals can be confined to a very small space (Fig. 2a), which facilitates ultra-high-precision additive manufacturing.*” (**Page 6-7, highlighted in blue**)

In addition to diffusion, it is clear that the nonlinear absorption index (N value) and shrinkage also have an effect on the precision of MOS, so we must discuss them objectively in order to give the reader a comprehensive understanding.

Thanks again for your comments.

14. The ordering of panels in Figure 3 is hard to follow. a-e are read left from right, but then f-p switch to up and down. Please be consistent within the figure.

Response:

We appreciate the comments. We have rearranged the order of panels in **Fig. 3**. Also, since **Fig. 3e-m** belongs to a set of data, we have separated them with dashed line for ease of reading.

15. The section title “An attempt to develop ZnO UV photodetector” is strange. It seems like the authors succeeded so why call it an “attempt”?

Response:

We appreciate the comments. We apologize for the confusing title and we have re-named this section title to be “Photodetectors fabricated by our strategy and the performance improvement”.

Notably, the superior performance of our devices can be clearly demonstrated by comparison with similar literature (**Supplementary Table 4**). Meanwhile, the PDCR of ZnO UV photodetector might be further improved by our strategy to reduce the line-width of ZnO (**Supplementary Fig. 19**).

16. It seems strange to call Figure 3o&p a 3D structure. In the context of multiphoton lithography, this would only be 2.5D. Do the authors try to print truly 3D structures with maybe only one resist as multiphoton lithography should be capable of this?

Response:

We appreciate the comments. As we mentioned above, we have supplemented the experiment, and provided several truly 3D structures (**Supplementary Fig. 15**) in the revised manuscript, demonstrating the 3D fabricating ability of our strategy. Since our original attention is on the ability of high-precision ROS fabrication, and 3D fabrication was overlooked in the previous manuscript. For 3D structures fabrication, the photoresist should be prepared with a high content (see **Methods section**), and drop-casted onto a glass substrate to achieve enough film thickness. We have made a brief

discussion about this issue in the revised manuscript (**Page 12, highlighted in blue**). For Figure 3o&p, the expression of “3D structure” have been removed, as shown below:

“...a three-layer multi-MOS consisting of ZnO, CuO and ZrO₂ is also successfully implemented (Fig. 3o-p) which means that this strategy has the potential to achieve the layer-by-layer fabrication of heterogeneous MOS.”

17. The authors only provide a very broad range of printing powers and speeds. The authors should be more specific and include the conditions of printing for each of the presented results.

Response:

We appreciate the comments. As requested, we have provided the specific parameters (in Figure or Figure caption or experimental section) for each of the presented results.

18. The pyrolysis temperatures used in this work are very large. Many other components of electrical circuits may not be compatible with these temperatures. Can the authors comment on this?

Response:

We appreciate the comments. This is a very good question. For all three metal oxides we have mentioned, a minimum sintering temperature of 550 °C is required. Sintering at 800 °C and 1000 °C is only used to study the crystalline phase transition process of ZrO₂. Obviously, many electronic circuit units cannot withstand the high temperature of 550 °C. This is the downside of sintering technology.

One alternative is to print components and then assemble them.

If it still doesn't work, you can consider the following aspects, which is what we are doing.

1) Using “metal oxide nanoclusters” as precursor photoresists, thereby reducing the sintering temperature. For example, Bauer et al. used a hybrid organic-inorganic polymer resin (POSS) as a feedstock material that requires a much lower temperature to achieve SiO₂ micro structures (**Fig. R10**). POSS

resin itself constitutes a continuous Si-O-Si network that forms transparent fused silica at only 650°C. This temperature is 500°C lower than the sintering temperatures for fusing discrete silica particles (J. Bauer et al. A sinterless, low-temperature route to 3D print nanoscale optical-grade glass. *Science* 380, 960-966 (2023). DOI:10.1126/science.abq3037). The preparation of universal “metal oxide nanoclusters” is a challenge, and perhaps experience can be gained from the methods used to prepare metal-organic framework materials.

[Figure redacted]

Fig. R10 Fabrication of high-quality fused silica nanostructures from an acrylate-functionalized POSS resin. (*Science*, 2023, 380, 960-966)

2) In addition to sintering, directional laser ablation should be an alternative method to remove organic matter from photoresists. The precursor photoresist pattern can be selectively sintered by laser without causing damage to other surrounding structures. This is a strategy we are trying, and the difficulty lies in inefficiency and high-precision alignment.

We have added a statement in the revised manuscript for this issue (**Page 15, highlighted in blue**), as shown below:

“.....Noted that many other components on the target device may not be compatible with the sintering temperature of MOS, one alternative is to print components and then assemble them.....”

19. The authors compare their results to the optical diffraction limit. However, this is confusing because they compare after pyrolysis results to techniques that do not involve any separate step after exposure and development. Their

comparison would make more sense if they compare the dimension of their prints before pyrolysis. I am dubious that they beat the STED technique before pyrolysis. To be clear, I am not saying that the authors shouldn't compare their final results to the final results for other techniques. But it seems suspect to frame their argument in comparison to techniques that do not do post-processing steps. Besides, couldn't STED be applied to their resist system leading to even smaller CD (considering the photoinitiator is DETC)?

Response:

We appreciate the comments and constructive advice.

Firstly, we must state that all the techniques in **Supplementary Table 2** and **Fig. 2i** all involve additional sintering after development, including EUV and EBL. They are all additive manufacturing + sintering strategies to prepare MOS, not just photoresist after development.

[Figure redacted]

Fig. R11 The ZnO lines made with EBL after heat treatment at 500 °C (left) and 700 °C (right). (Adv, Mater, 005, 17, 1757-1761)

For example, in "Saifullah, M.S.M., et al., Sub-10 nm High-Aspect-Ratio Patterning of ZnO Using an Electron Beam. *Advanced Materials*, 2005, 17: 1757-1761.", they used EBL to exposure metal naphthenate, and the developed lines were treated at 500 °C and 700 °C for 1h to remove the organic matter, as shown in **Fig. R11**. Likewise, in "Auzelyte, V., et al., Direct formation of ZnO nanostructures by chemical solution deposition and EUV exposure. *Nanotechnology*, 2010, 21: 215302.", they also used EUV to exposure metal naphthenate, and the developed lines were treated at 850°C to realize ZnO pattern (**Fig. R12**).

[Figure redacted]

Fig. R12 Zinc naphthenate nanostructures made with EUV-IL (left) and corresponding ZnO nanostructures formed after the heating at 850 °C (right). (Nanotechnology, 2010, 21: 215302)

Indeed, our Figure caption (**Fig. 2i**) and Table caption (**Supplementary Table 2**) have clearly shown that it is a comparison of MOS prepared by various strategies than the photoresist after development.

Since MOS is sintered from precursor photoresist, our precursor photoresist has different sintering shrinkage than the comparative literature, so it might be more appropriate to compare the CD of the final MOS rather than photoresist; after all, we are not looking to utilize the precursor photoresist for pattern transfer.

Secondly, we can not beat STED before sintering. We have previously observed the minimum width of the lines (Zn-based precursor photoresist) before sintering, which was about 45 nm (**Fig. R13**).

Fig. R13 SEM of the minimum width of the lines (Zn-based precursor photoresist) before sintering.

In recent years, Martin Wegener, Joachim Fischer and et al, have done a lot of research on STED lithography [**Ref. 1-5 below**], which is very admirable. Following them, our project team has also carried out some research on STED

lithography, and the CD has reached below 40 nm on a substrate [Ref. 6-12 below].

Thirdly, STED can not be applied to our photoresists because the unknown reason, even though we use DETC as initiator. As we mentioned in your comment **13#**, we found that our photoresists lost STED (stimulated emission dissipation) capability, because there may be an energy transfer between metal complexes and DETC (our presumption). In details, by absorbing a femtosecond laser, the metal complexes in the excited state might be able to transfer energy to the ground state DETC (S_0), and then the excited DETC (S_n) undergoes intersystem crossing to produce free radicals. This process may lead to STED failure, and this is why we didn't use STED lithography to achieve a smaller CD.

We are investigating the underlying reason of STED failures, which can not only help us to understand the STED process more comprehensively, but also provide guidance for the design of new metal-complex resins / initiators. We hope we can realize STED lithography for metal-complex to obtain a smaller CD in the future.

Refs.

1. Fischer, J., et al., Exploring the Mechanisms in STED-Enhanced Direct Laser Writing. *Advanced Optical Materials*, 2015. 3(2): p. 221-232.
2. Fischer, J., G. von Freymann, and M. Wegener, The Materials Challenge in Diffraction-Unlimited Direct-Laser-Writing Optical Lithography. *Advanced Materials*, 2010. 22(32): p. 3578-3582.
3. Fischer, J. and M. Wegener, Three-dimensional direct laser writing inspired by stimulated-emission-depletion microscopy. *Optical Materials Express*, 2011. 1(4): p. 614-624.
4. Fischer, J. and M. Wegener, Ultrafast Polymerization Inhibition by Stimulated Emission Depletion for Three-dimensional Nanolithography. *Advanced Materials*, 2012. 24(10): p. OP65-OP69.
5. Fischer, J. and M. Wegener, Three-dimensional optical laser lithography beyond the diffraction limit. *Laser & Photonics Reviews*, 2013. 7(1): p. 22-44.
6. Cao, C., et al., Dip-In Photoresist for Photoinhibited Two-Photon Lithography

to Realize High-Precision Direct Laser Writing on Wafer. ACS Applied Materials & Interfaces, 2022. 14(27): p. 31332-31342.

7. Dazhao, Z., et al., Direct laser writing breaking diffraction barrier based on two-focus parallel peripheral-photoinhibition lithography. Advanced Photonics, 2022. 4(6): p. 066002.

8. Ding, C., et al., Subdiffraction 3D Nanolithography by Two-Photon Two-Step Absorption and Photoinhibition. Laser & Photonics Reviews, 2024. 18(3): p. 2300645.

9. He, M., et al., Single-color peripheral photoinhibition lithography of nanophotonic structures. PhotonIX, 2022. 3(1): p. 25.

10. Qiu, Y., et al., Peripheral-photoinhibition-based direct laser writing with isotropic 30 nm feature size using a pseudo 3D hollow focus. Optics & Laser Technology, 2024. 170: p. 110011.

11. Su, C., et al., Sub-diffraction optical beam lithography based on a center-non-zero depletion laser. Optics Letters, 2024. 49(1): p. 109-112.

12. Guan, L., et al., Light and matter co-confined multi-photon lithography. Nature Communications, 2024. 15(1): 2387.

20. The labeling of scalebars in many of the images is missing.

Response:

We appreciate the comments. We have provided the labeling of scale bars, some in Figures and some the Figure captions.

21. What is the spacing of the ZnO UV imaging array? Should provide more info than just 10 x 10.

Response:

We appreciate the comments. We have provided detailed parameters of the ZnO UV sensor preparation (including unit and imaging arrays) in the experimental section (**Page 17, highlighted in blue**).

Reviewer #3 (Remarks to the Author):

The research manuscript is valuable and timely, has significant advancements in the field, but the description is too generic, pompous, and vague due to lack of support.

Response:

We appreciate your positive comments and useful suggestions, which certainly help us to improve the quality of our manuscript.. In response to your concerns, we have performed additional experiments and revised the presentation to make our manuscript more convincing.

1. TITLE: What is “ultra-high precision”? – it’s a vague and not supported.

Response:

We appreciate the comments. Obviously, "ultra-high precision" is a limiting "adverb" or "adjective" that in our context is limited to describing the fields of “additive manufacturing of metal oxide semiconductors via multiphoton lithography”.

First of all, ultra-high precision is relative. Compared with the previous MOS fabricated by additive manufacturing and MPL technology, we prepared MOS with a precision of 35 nm, setting a benchmark in this area. As shown in **Fig. 2i** and **Supplementary Table 2**, we have provided comparative data. Secondly, ultra-high precision is also absolute. In fact, 35 nm has reached the level of DUV lithography. Therefore, we believe that it is reasonable and fair to use the expression “ultra-high precision” in title.

Notably, the title has been modified to be “Ultra-high Precision Nano Additive Manufacturing of Metal Oxide Semiconductors Via Multi-photon Lithography”, as recommended by you.

Fig. 2i Comparison of critical dimension of MOS additive manufacturing technologies. Data and references are listed in **Supplementary Table 2**.

Supplementary Table 2 Comparison of minimum characteristic dimension of metal oxides additive manufacturing technology.

#	Groups	Technologies	Materials	CD (μm)	Ref
1	Farandos et al., 2016	Inkjet printing	Yttria-stabilized zirconia (YSZ)	35	[4]
2	Li et al., 2022	Digital light process	CuO, NiO, etc.	28	[5]
3	Kim et al., 2016	Electrohydrodynamic Inkjet Printing	In ₂ O ₃	2	[6]
4	Zhang et al., 1999	Micro-stereolithography	Al ₂ O ₃	1.2	[7]
5	Yu et al., 2018	Two-Photon lithography (TPL)	TiO ₂	0.65	[8]
6	Passinger et al., 2007	TPL	TiO ₂	0.4	[9]
7	Cho et al., 2020	area-selective atomic layer deposition	ZnO, Al ₂ O ₃ , SnO ₂	0.312	[10]
8	Yang et al., 2023	Laser-induced hydrothermal synthesis	ZnO	0.26	[11]
9	Yee et al., 2019	TPL	ZnO	0.25	[12]
10	Long et al., 2020	TPL	ZnO (nanowires)	0.24	[13]
11	Liu et al., 2021	TPL	ZnO, Co ₃ O ₄	0.17	[14]
12	Guo et al., 2010	TPL	SnO ₂	0.15	[15]
13	Gailevičius et al., 2018	TPL	ZrO ₂	0.1	[16]
14	Desponds et al., 2021	TPL	ZrO ₂	0.1	[17]
15	Malinauskas et al., 2022	TPL	SiO ₂ /ZrO ₂	0.06	[18]
16	This work	MPL	ZnO, CuO, ZrO ₂	0.035	
17	Auzelyte et al., 2010	EUV	ZnO	0.01	[19]
18	Saifullah et al., 2005	EBL	ZnO	0.005	[20]

2. What is “metal oxides semiconductor?” – plural to singular material/device?

Response:

We appreciate the comments. It should be “metal oxide semiconductors”. What we want to express is that a wide range of metal oxide based semiconductors can be prepared by our method. We apologize for this mistake.

3.MPL, which is the employed technique could be added to the title making it more informative of what technique was used instead of hollow calling it “ultra-high precision”.

Response:

We appreciate the comments. This is a very good suggestion. As recommended, we have revised our title to make our title more informative and specific.

4.“Typically, we fabricated ZnO photodetector..” – typically fabricated? What does it mean. And in the main text it is explained as an attempt to fabricate, while in the Discussion it converts to fabricated photodetectors. So was it one, many, or just an attempt to make it?

Response:

We appreciate the comments. We apologize that our previous statement may be inaccurate and confusing for you. We initially wanted to express that we had only provided the reader with one sample of device preparation (ZnO photodetector), and that there were many other devices that we could not validate individually, hence the use of the expression of "typically" and “an attempt to fabricate”. We have replaced "typically" by “for instance”. Besides, we have modified “an attempt to fabricate” to be “photodetectors fabricated by our strategy and the performance improvement”, since we did succeed in preparing ZnO photodetector units and arrays. Notably, the superior performance of our devices can be clearly demonstrated by comparison with similar literature (**Supplementary Table 4**).

5. ABSTRACT: Why specifically 35 nm are referred as critical dimensions (CD)? It is not supported later in the manuscript either.

Response:

We appreciate the comments. Critical dimensions (CD) is a technical term used in lithography to refer to the size of the smallest structure that can be obtained [Ref.1-3 below]. It generally refers to the minimum line width, and also named as “half-pitch” or “feature size” [Ref.4 below].

Since ZnO exhibited the minimum line width of 35 nm (Fig. 2f), as we discussed in the section of “Ultra-high precision nano additive manufacturing of MOS via MPL”, therefore, we believe it is reasonable to refer 35 nm as CD in the abstract.

Fig. 2f the ZnO lines fabricated by our strategy.

Refs.

1. Timko, A.G., et al., Linewidth reduction using liquid ashing for sub-100 nm critical dimensions with 248 nm lithography. *Journal of Vacuum Science & Technology B: Microelectronics and Nanometer Structures Processing, Measurement, and Phenomena*, 2001. 19(6): p. 2713-2716.
2. Guo, L., X. Wang, and H. Huang, Analysis of illumination pupil filling ellipticity for critical dimensions control in photolithography. *Chinese Optics Letters*, 2006. 4(4): p. 237-239.
3. GENG, Y., Real-time monitoring and control of critical dimensions in Lithography. ScholarBank@NUS Repository, 2012.
4. Gan, Z., et al., Three-dimensional deep sub-diffraction optical beam lithography with 9 nm feature size. *Nature Communications*, 2013, 4, 2061.

6. INTRODUCTION: CD are mentioned to be of struggle reaching below 60 nm, yet have a look at recent related publication on that matter, which also employs solid MOS photoresist and achieves 60 nm: Laser additive manufacturing of Si/ZrO₂ tunable crystalline phase 3D nanostructures. Opto-Electron Adv 5, 210077 (2022); <https://doi.org/10.29026/oea.2022.210077>

Response:

We appreciate the comments. We have carefully read the literature you provided and it is a very impressive work. Its realized Si/ZrO₂ tunable crystalline phase 3D nanostructures using femtosecond laser and the CD of Si/ZrO₂ lines below 60 nm.

We've changed the statement, that is "...CD is hardly less than 60 nm..."(**Page 3, highlighted in blue**), and cited this article in the revised manuscript (**Page 20, highlighted in blue**).

[Figure redacted]

Fig. R14 3D woodpiles from Opto-Electron Adv 5, 210077 (2022).

However, there is actually a big difference between this article and our work. Their strategy is suitable for the processing of micro- and nano-3D devices, but it might be difficult to realize ultra-high-precision structures on substrates, especially lines. Their CD (sub 60 nm) is realized by means of fabricating 3D woodpiles (**Fig. R14**), in which the lines are not restricted by the substrate and are more like free suspension lines, which will become thinner by tension during the MPL process, such as 9 nm reported by prof. Xuanming Duan [**Ref.1 below**]. Recently we have carried out a similar study and found that free suspension lines are easily obtained below 30 nm by MPL [**Ref.2-3 below**].

Similarly, this kind of lines, which are not restricted by the substrate, will have higher shrinkage rate during sintering, and thus higher precision can be realized. However, the disadvantages of this strategy are very obvious. The tensile shrinkage and sintering shrinkage of the suspension line (or lines in 3D woodpiles) in MPL are difficult to be controlled stably, which can easily lead to defects such as twisted lines, broken lines, large edge roughness, etc., and the

line width is poorly homogeneous (**Fig. R14**, 99-129 nm at same condition), which makes it difficult to repeat the processing.

In addition, suspended micro-structures can generally only be used for 3D micro-structures such as photonic crystals, while many devices need to be processed on a substrate (layer by layer). We suspect that the strategy of the above article will be substantially weakened in terms of MOS precision once it is used for fabricating on substrates.

Refs.

1. Gan, Z., et al., Three-dimensional deep sub-diffraction optical beam lithography with 9 nm feature size. *Nature Communications*, 2013, 4, 2061.
2. Cao, C., et al., High-Precision and Rapid Direct Laser Writing Using a Liquid Two-Photon Polymerization Initiator. *ACS Applied Materials & Interfaces*, 2023. 15(25): p. 30870-30879.
3. Qiu, Y., et al., Peripheral-photoinhibition-based direct laser writing with isotropic 30 nm feature size using a pseudo 3D hollow focus. *Optics & Laser Technology*, 2024. 170: p. 110011.

7. "Atomically economical.."- what does it stands for?

Response:

We appreciate the comments. Atomically economical means "atom economy". It is an important concept in green chemistry. Atom economy is the conversion efficiency of a chemical process in terms of all atoms involved (desired products produced). The ideal atom economy reaction is one in which 100 percent of the atoms in the raw material molecules are converted into products, with no by-products or waste.

Obviously, as on kind of additive manufacturing technology, MPL is a more atomically economical strategy than traditional piece-reduced manufacturing and pattern-transfer strategy.

8. The mentioned limited Library of photoresists for MPL combined with pyrolysis is actually has been provided in a recent perspective paper at Fabrication of Glass-Ceramic 3D Micro-Optics by Combining Laser Lithography

and Calcination, Adv. Func. Matter., 2215230 (2023); <https://doi.org/10.1002/adfm.202215230>.

Response:

We appreciate the comments and recommendation. We have read the paper carefully, and it reports on 3D printing of glass-ceramic-based micro-optics by femtosecond laser and pyrolysis.

We have cited this article in the INTRODUCTION section of the revised manuscript (**Page 19, highlighted in blue**), and it will make our article more comprehensive for readers.

As we mentioned above (response to your **comment #6**), the biggest difference between this article and our work is that we focus more on how to achieve high precision on the surface of the substrate rather than micro 3D structures.

9. Among Bauer and Xiong, there was also Gonzalez, who fabricated crystalline micro-optics using very similar photoresist, exposure and post-processing strategy, should be included in overview: Laser 3D Printing of Inorganic Free-Form Micro-Optics, Photonics 8, 577 (2021); <https://doi.org/10.3390/photonics8120577>

Response:

We appreciate the comments and recommendation. We have introduced this work in INTRODUCTION section, and also cited it in the revised manuscript. (**Page 20, highlighted in blue**)

10. RESULTS: “Green laser..” – was it the color of the laser packaging box? Anyhow, “green” lasers have been successfully exploited to fabricate via MPL means even not photo-sensitized resins, see: Femtosecond-Laser Direct Writing 3D Micro-/Nano- Lithography Using VIS-Light Oscillator, J. Centr. South Univ., 29, 3270-3276 (2022); doi: 10.1007/s11771-022-5153-z.

Response:

We appreciate the comments.

It is not the color of the laser packaging box, but the color of laser, since the wavelength of the used laser is 525 nm. Of course, green lasers have been used many times in MPL, and the article you mentioned used a 517 nm green laser for two-photon processing.

In the past few years, we mainly used 780 nm femtosecond lasers for multiphoton lithography, but since the diffraction limit is proportional to the wavelength, it is difficult to improve the processing precision. Therefore, recently we prefer to use short-wave femtosecond lasers to improve the fabrication precision. However, the shorter the laser wavelength, its penetration in the photoresist may decrease, resulting in the inability to perform MPL. Therefore, we provide the transmission spectra of the photoresist film in the manuscript to demonstrate that our photoresist material has good penetration at 525 nm wavelength and can be processed without any problem.

Our previous statement may not have been clear enough, so we have modified it in the revised manuscript (**Page 4, highlighted in blue**), as seen below:

“Although the resultant photoresist films inherit the color of the metal ions and DETC (Supplementary Fig. S2), they still have good transmittance to the wavelength (525 nm) of the used laser (Supplementary Fig. 3-5), which can ensure the feasibility of MPL.”

Besides, the mentioned literature “Femtosecond-Laser Direct Writing 3D Micro-/Nano- Lithography Using VIS-Light Oscillator, J. Centr. South Univ., 29, 3270-3276 (2022)” has been cited in the revised manuscript (**Page 18, highlighted in blue**) to make our work more convincing.

11. There is not 3D structure demonstrated, though it is mentioned as the MPL is advantageous for it. Why? Previous pointed publications available at <https://doi.org/10.29026/oea.2022.210077> and <https://doi.org/10.3390/photonics8120577> clearly demonstrate the feasibility of the method to construct 3D architectures of complex geometries and high resolution.

Response:

We appreciate the valuable comments. we have supplemented the experiment, and provided several 3D structures (**Supplementary Fig. 15**) in the revised manuscript, demonstrating the 3D fabricating ability of our strategy. Since our original attention is on the ability of high-precision ROS fabrication, and 3D fabrication was overlooked in the previous manuscript. For 3D structures fabrication, the photoresist should be prepared with a high content (see **Methods section**) and drop-casted onto a glass substrate to achieve enough film thickness.

We have made a brief discussion about this issue in the revised manuscript (**Page 12, highlighted in blue**).

Supplementary Fig. 15 3D micro-structures fabricated by our strategy using zirconium-based photoresist. **a** Nut array (the thread on the side can be clearly seen.) and **b** diamond cubic lattice structure.

12. What are the photo structuring mechanisms? It is shown in some details, but gives an ambiguous impression. Fig 2 (b, c, and d) shows line width dependence on laser power (P), in (g) it is plotted at a log-log scale and in (h) explained as intensity ^{N} (I^N) law, but in the caption it is mentioned as threshold dose (D or E). So which is the determining parameter and how they are (inter-)related? There is a recent paper studying mechanisms for various wavelengths and pulse durations, revealing the mechanisms contributing to voxel growth, should be discussed and compared: X-photon laser direct write 3D nanolithography, *Virt. Phys. Prototyp.* 18, e2228324 (2023); <https://doi.org/10.1080/17452759.2023.2228324>

Response:

We appreciate the comments. Please note that **Fig. 2g** and **Fig. 2h** are interchanged in the revised manuscript, as recommended by other reviewer.

Section A: Photo structuring mechanism

In term of photo structuring mechanisms (photo induced radical polymerization), it has been well studied, both in the field of MPL (**Refs.1-4 below**) and in the field of polymer polymerization [**Refs.5-6 below**]. Thus we have previously given neither a schematic nor a description in the previous manuscript. We apologize for the confusion this may have caused some readers, so we have added some explanation in the revised manuscript (**Page 4, highlighted in blue**), as seen below:

“During the MPL process, DETC in the exposure area can generate a large number of free radicals through multi-photon absorption, which in turn triggers the polymerization of the carbon-carbon double bonds in the precursor photoresist, leading to a significant solubility difference from the unexposed photoresist, and ultimately the target pattern is obtained after development.”

Section B: Explanation of Fig.2

We first given the relationship between the line-widths and laser power in **Fig.2b-d**. Due to the threshold effect [**Refs.7 below**], the linewidth increases with the laser power, which is similar to that in many articles [**Refs.8-10 below**].

Next in **Fig.2g** and **Fig.2h**, we are trying to investigate why the minimum line width of the three metal oxides is different (ZnO (35 nm) < CuO (36 nm) < ZrO₂ (58 nm)). It is believed that the minimum line width depends on the nonlinearity absorption exponent (**N**) of the photoresist, apart from wavelength (λ), numerical aperture (NA), and technical factor (k) [**Refs.11-15 below**], and the larger the **N**, the more favorable to achieve a high precision. The **N** can be obtained by plotting a linear fit between exposure time (t_{exp}) and the threshold power (P_{th} , at the corresponding exposure time) through a double-logarithmic representation [**Refs.16-19 below**], and calculating from the slope of the linear fit line (**N=-1/slope**), e.g., **Fig. 2h**.

Fig. 2g shown a model that used to qualitatively calculate the theoretical minimum line-widths that can be achieved with different **N** values (one-photon, two-photon and three-photon), where the horizontal coordinate was the distance (or cross-section of the laser spot within the photoresist) and the

vertical coordinate represented the relative intensity distribution of the laser (within the photoresist cross-section). This is combined with the real **N** obtained in **Fig. 2h** to illustrate why the three metal oxides have different minimum line-widths.

Fig. 2b-d were performed at a fixed writing speed, and just varying the laser power. But for **Fig. 2h**, a line arrays that with a wide range of laser power and writing speed (exposure time) should be fabricated at the same time. And seeking the threshold power (**P_{th}**) at each exposure time (**t_{exp}**). For **Fig. 2**, the exposure time is a constant, so it makes sense that the vertical coordinate is labeled as intensity, since threshold dose = threshold intensity × exposure time. Similarly, as shown in **Fig. R15**, Gan ZS et al. employed a model similar to ours to predict the line-width [**Ref. 7 below**].

[Figure redacted]

Fig. R15 the model used for predicting line-width (Nature Communications, 2013. 4. 2061).

As a result, Zn-based precursor photoresist has the highest **N** (3.76), followed by Cu-based precursor photoresist (**N** = 3.52) and Zr-based precursor photoresist (**N** = 2.76), which leads to the order of their CD (ZnO (35 nm) < CuO (36 nm) < ZrO₂ (58 nm)).

We have provided detailed explanation and discussion in the manuscript. (**Page 8-9, highlighted in blue**)

[Figure redacted]

Fig. R16 Lateral voxel size evolution with revealed global slope γ change tendencies. The fit-obtained N values are shown at the corresponding lines. (X-photon laser direct write 3D nanolithography, *Virt. Phys. Prototyp.* 2023,18, e2228324)

We appreciate for providing the mentioned paper, which is very impressive. It reported 3D nanolithography with ultra-short ≈ 100 fs pulses at a wide visible-to-near-IR spectral range of 400–1200 nm ($N=0.8 - 4.6$). As shown in **Fig. R16**, larger N can yield smaller line widths (320 nm for 4.6, and 360 nm for 0.9 nm), which is in agreement with our results.

We have cited this paper in the revised manuscript to make our work more convincing. (**Page 21, highlighted in blue**)

Refs.

1. Fischer, J., G. von Freymann, and M. Wegener, The Materials Challenge in Diffraction-Unlimited Direct-Laser-Writing Optical Lithography. *Advanced Materials*, 2010. 22(32): p. 3578-3582.
2. Fischer, J. and M. Wegener, Three-dimensional direct laser writing inspired by stimulated-emission-depletion microscopy. *Optical Materials Express*, 2011. 1(4): p. 614-624.
3. Guan, L., et al., Light and matter co-confined multi-photon lithography. *Nature Communications*, 2024. 15(1): p. 2387.
4. Tang, J. et al. Ketocoumarin-based photoinitiators for high-sensitivity two-photon lithography. *ACS Appl. Polym. Mater.* 5, 2956–2963 (2023)
5. Ishizu, K. and A. Mori, Synthesis of hyperbranched polymers by self-

addition free radical vinyl polymerization of photo functional styrene. *Macromolecular Rapid Communications*, 2000. 21(10): p. 665-668.

6. Moszner, N., et al., Monomers for adhesive polymers, 2. Synthesis and radical polymerisation of hydrolytically stable acrylic phosphonic acids. *Macromolecular Chemistry and Physics*, 1999. 200(5): p. 1062-1067.

7. Gan, Z., et al., Three-dimensional deep sub-diffraction optical beam lithography with 9 nm feature size. *Nature Communications*, 2013. 4. 2061

8. Cao, C., et al., Cellulose derivative for biodegradable and large-scalable 2D nano additive manufacturing. *Additive Manufacturing*, 2023. 74: p. 103740.

9. Cao, C., et al., Click chemistry assisted organic-inorganic hybrid photoresist for ultra-fast two-photon lithography. *Additive Manufacturing*, 2022. 51: p. 102658.

10. Cao, C., et al., High-Precision and Rapid Direct Laser Writing Using a Liquid Two-Photon Polymerization Initiator. *ACS Applied Materials & Interfaces*, 2023. 15(25): p. 30870-30879.

11. X. Zhou, Y. Hou, J. Lin, *AIP Advances* 2015, 5.

12. B. Fay, *Microelectronic Engineering* 2002, 61-62, 11.

13. K.-S. Lee, R. H. Kim, D.-Y. Yang, S. H. Park, *Progress in Polymer Science* 2008, 33, 631.

14. M. Rothschild, *Materials Today* 2005, 8, 18.

15. M. Malinauskas, M. Farsari, A. Piskarskas, S. Juodkazis, *Physics Reports* 2013, 533, 1.

16. Hahn, V., et al., Two-step absorption instead of two-photon absorption in 3D nanoprinting. *Nature Photonics*, 2021. 15(12): p. 932-938.

17. Fischer, J.; Mueller, J. B.; Kaschke, J.; Wolf, T. J.; Unterreiner, A. N.; Wegener, M. Three-dimensional multi-photon direct laser writing with variable repetition rate. *Opt Express* 2013, 21, 26244-26260

18. Yang, L.; Münchinger, A.; Kadic, M.; Hahn, V.; Mayer, F.; Blasco, E.; Barner-Kowollik, C.; Wegener, M. On the Schwarzschild Effect in 3D Two-Photon Laser Lithography. *Advanced Optical Materials* 2019, 7, 1901040.

19. Ding, C., et al., Subdiffraction 3D Nanolithography by Two-Photon Two-Step Absorption and Photoinhibition. *Laser & Photonics Reviews*, 2024. 18(3): p. 2300645.

13. There is no image of the real fabricated photodetector in Fig. 4. And it is not clear of what dimensions were used and if the critical 35 nm was necessary or beneficial to achieve it.

Response:

We appreciate the comments.

Section A: About the real fabricated photodetector

Instead of making a complete UV detector (including sensor unit, signal receiver, processor and display), which would have been unnecessary and pointless for us, we simply prepared the sensor unit (**Fig. 4b**) and tested its sensing performance using probes (**Fig. R17**).

In fact, many reports [**Refs.1-2 below and Supplementary Table 4**] that specialize in preparing detectors have done the same as ours, e.g., Liu, X., et al. reported a all-printable band-edge modulated ZnO nanowire photodetectors (**Fig. R18**), which also contains only the sensing unit. Thus, this may be a reasonable and widely accepted method. We believe this is sufficient proof of the capability of the strategy we offer.

Fig. R17 Testing device sensing performance with probes.

[Figure redacted]

Fig. R18 All-printable band-edge modulated ZnO nanowire photodetectors. (Nature Communications, 2014. 5(1): p. 4007)

Section B: The dimensions of photodetector unit

We have added the scale bar in **Fig. 4b**, while the exact dimensions and preparation of the sensor are described in the experimental section. The length and width of the rectangular zinc oxide in the sensor unit are 100 μm and 20 μm , respectively. The length and width of ITO electrodes are 100 μm and 100 μm , and the spacing is 40 μm .

Section C: Response to “whether the critical 35 nm was necessary or beneficial for photodetector”

We had not provided the photodetector with 35 nm ZnO because the reason we discussed below, but we have demonstrated that high-precision lines is benefit to achieve photodetector.

We have explored the effect of line width on performance (**Supplementary Fig. 19**), and made a comprehensive explanation below.

As can be seen from **Supplementary Fig. 19**, while keeping the number of lines constant, as the line-width becomes smaller, the photo-to-dark current ratio (PDCR) increases dramatically. PDCR represents the signal-to-noise ratio, which is a key metric for sensors, thus illustrating that a high precision ROS is helpful in improving performance. Obviously, this performance improvement may also occur in other micro-devices.

Since it is difficult to process 35 nm ROS on a silicon substrate using an air objective with a low numerical aperture (NA0.95), as mentioned by reviewer #1, and our oil objective (NA1.45) is not suitable for silicon substrate. So here we only provide linewidths from 0.5 μm to 2 μm , but its enough to illustrate the beneficial effect of high precision on performance. Also, after consulting with optical experts, once the line width is less than 1/4 of the wavelength (365 nm), it may be difficult for the ROS pattern to absorb photons efficiently. However, this does not mean that ultra-high precision ROS cannot be used in other miniature devices, especially microelectronic devices.

Although high precision is not necessary for some devices, if high precision is possible, it is obviously benefit to device miniaturization and integration, which is the trend of the future. This can not be completed by the processing method in **Supplementary Table 2**.

Supplementary Fig. 19 Optical microscope images (up: open-field mode, down: dark-field mode) of ZnO UV photodetector with different line-widths: **a** 0.5 μm , **b** 1 μm , **c** 2 μm . I–V characteristic curves of ZnO UV photodetector in the dark and under 365 nm UV illumination (0.1 mW cm^{-2}): **d** 0.5 μm , **e** 1 μm , **f** 2 μm .

Obviously, the realization of ultra-high-precision MOS is just one of our important contributions, but not all. The ZnO UV photodetectors prepared by our strategy have superior performance. To demonstrate the advancement of our devices, we have listed the relevant literature for comparison in the revised manuscript (**Supplementary Table 4**, as shown below). The superior performance of our devices can be clearly evidenced.

The possible reasons for the superior performance of the ZnO UV photodetectors prepared by our strategy are as follows:

The photoresist we use has a high metal content (**Supplementary Table 1**), and the resulting ROS microstructures have fewer defects (holes, discontinuities, low crystallinity, **Fig. 1-2**), which allows for higher performance. For example, we have prepared ZnO photodetectors with PDCR value up to 10500 (365 nm,) and 100000 (254 nm), respectively (**Supplementary Fig.**

S16a, Fig. 4e and Supplementary Table 4), which are higher than the majority of pure ZnO photodetectors (including various structures) (**Supplementary Table S4**).

Meanwhile, homogeneous elemental doping (atomic level) can be easily achieved within our photoresist systems. For example, aluminum acrylate is mixed into the photoresist, and aluminum acrylate acts as a reactive monomer during the polymerization process and participates in the construction of a robust polymer network (**Supplementary Fig. S12a**), ensuring that the Al atoms are not lost / aggregate during the development / sintering process. This can facilitate the performance of the element-doped devices. Consequently, remarkable performance can be obtained by our strategy, e.g., the PDCR of Al-doped ZnO photodetector reaches to 16000 (10500 for pure ZnO photodetector) at 365 nm, which are higher than the element-doped ZnO photodetector(**Supplementary Table 4**).

The ZnO photodetectors prepared by the existing techniques in **Supplementary Table 2** and **Table S4** do not have the above mentioned advantages, and hence their performance is hardly superior to ours.

Supplementary Table 4 The performance comparison of ZnO UV photodetectors.

Authors	Types of MOS	PDCR value	Journal, year	Ref.
Basavaraj G. Hunashimarad, et al.	Ca-doped ZnO film	3.17 at 365 nm	Optical Materials 2022	[1]
Sabina M. Hatch, et al.	ZnO-nanorods-CuSCN	4.5 at 375 nm	Advanced materials 2013	[2]
Shoou-Jinn Chang, et al.	Fe-doped ZnO	<10 at 375 nm	IEEE Photonics Technology Let. 2013	[3]
Qi Li, et al.	ZnO-CuO nanorod	10 at 325 nm	Advanced optical materials 2022	[4]
Chih-Hung Hsiao, et al.	Needle-like Ga-ZnO nanorods	11.07 at 360 nm	IEEE transactions on electron devices 2013	[5]
Jing Wang, et al.	ZnO nanowires	24.2 at 365 nm	J. Mater. Chem. C 2016	[6]
Chiung-Hsien Huang, et al.	Li-doped ZnO nanorods	34.87 at 380 nm	Microsystem Technologies 2022	[7]
Sunghoon Park, et al.	ZnO nanowires	49 at 365 nm	Journal of Alloys and Compounds 2016	[8]
Min Chen, et al.	ZnO hollow-sphere nanofilm	53 at 350 nm	Small 2011	[9]
Huihui Yu, et al.	Atomic-thin ZnO Sheet	69.6 ay 365 nm 120.1 at 254 nm	Small 2020	[10]
Cheng-Liang, Hsu et al.	Vertical ZnO nanowires	67.5 at 254 nm	Chemical Physics Letters 2005	[11]

Soo Hyun Lee, et al.	ZnO nanorods	1720 at 380 nm	Nanoscale Research Letters 2016	[12]
Shaivalini Singh, et al.	Al-doped ZnO	3327.94 at 372 nm	MicrosystemTechnologies 2016	[13]
Nishant Kumar, et al.	Mg-doped ZnO films	71.68 at 365 nm	Journal of Alloys and Compounds 2018	[14]
Nishant Kumar, et al.	Cd-doped ZnO	93.78 at 386 nm	Journal of Alloys and Compounds 2017	[15]
S.J. Young, et al.	ZnO film	290 at 370 nm	Sensors and Actuators A: Physical 2007	[16]
Yen-Lin Chu, et al.	Ni-doped ZnO	393.04 at 380nm	J. Electrochem. Soc. 2020	[17]
Fatemeh Abbasi, et al.	Ni-doped ZnO film	416.14 at 350 nm	Optics Communications 2021	[18]
Ramazanali Dalvand, et al.	ZnO nanoneedles	600 at 325nm	Journal of Materials Science: Materials in Electronics 2018	[19]
Liu kw, et al.	Mg-doped ZnO film	Approx. 1000 at 368 nm	Sensors 2010	[20]
Akshta Rajan, et al.	ZnO thin film	1000 at 365 nm	MRS Online Proceedings Library 2013	[21]
Hsiang-Chun Wang, et al.	Ag nanoparticles modified ZnO	1000 at 365 nm	Nanoscale Research Letters 2020	[22]
Zeping Li, et al.	ZnO quantum dot	1767.8 at 350 nm	Applied Surface Science 2022	[23]
Jingwei Liu, et al.	ZnO nanowire	3000 at 365	Adv. Mater. Technol. 2022	[24]
James Taban Abdalla, et al.	ZnO nanorod	4000 at 365 nm	Journal of Electronic Materials 2020	[25]
Fa Cao, et al.	ZnO-CuI heterostructure	4250 at 365nm	Journal of Alloys and Compounds 2021	[26]
Dawit Gedamuv, et al.	ZnO nanotetrapod networks	4500 at 365 nm	Advanced Materials 2014	[27]
Omar F. Farhat, et al.	ZnO nanoaggregates	8345 at 365 nm	Sensors and Actuators A: Physical 2021	[28]
Amit Kumar Rana, et al.	Co3O4-ZnO film	45700 at 365 nm	Materials Science in Semiconductor Processing 2020	[29]
Jin Hyung Jun, et al.	ZnO nanoparticles	10 ⁶ at 325 nm	Ceramics International 2009	[30]
	ZnO wire	10500 at 365 nm	This work	
	Al-doped ZnO wire	16000 at 365 nm	This work	
	ZnO wire	100000 at 254 nm	This work	

Section D: Some necessary explanation

We want to emphasize that, our original purpose is to provide the readers or researchers a strategy that enables ultra-high-precision MOS fabrication, homogeneous elemental doping, heterogeneous structure preparation, and large area array preparation (the techniques described in **Supplementary Table 2** are difficult to realize or are not mentioned), giving readers a wider range of choices in their devices.

In fact, we would prefer to use our strategy to process a real integrated circuit chip unit, such as transistors, logic gates, etc., but this requires very demanding experimental conditions that we are not able to realize for the time being, so we only show ZnO sensors to illustrate the practicality of our strategy.

Thank you very much for your valuable suggestions and reminders. Supplementing the experiments and revising the manuscript as you requested not only further emphasizes the advancement of the manuscript, but also arouses the interest and attention of a wider audience.

We have added **Supplementary Fig. S19** and **Supplementary Table 4** into the “**Supplementary Information**” file, and have added some statement in the revised manuscript (**Page 13, highlighted in blue**).

Refs.

1. Liu, X., et al., All-printable band-edge modulated ZnO nanowire photodetectors with ultra-high detectivity. *Nature Communications*, 2014. 5(1): p. 4007.

2. Jin, Z., et al., Graphdiyne:ZnO Nanocomposites for High-Performance UV Photodetectors. *Advanced Materials*, 2016. 28(19): p. 3697-3702.

14. DISCUSSION: “new strategy” appears not to be new anymore once the pointed reports will be referenced, but rather an improvement of the previous works, which is substantial.

Response:

We appreciate the comments. To make our manuscript more rigorous, we revised the statement in manuscript. The word “new” has been removed.

15. REFERENCES: At least [2,4,7,8] have no page numbers.

Response:

We appreciate the comments. We have updated the references and added all omissions as you mentioned.

16. SUPPLEMENTARY: Table S3 should be updated with corresponding paper in <https://doi.org/10.29026/oea.2022.210077> benchmarking 60 nm linewidth in 3D nano-structure.

Response:

We appreciate the comments. As requested, we have added the mentioned paper in Table S3 (named **Supplementary Table 2** in the revised manuscript).

Point-to-point Response to Reviewers

Reviewer #1 (Remarks to the Author):

The author has already solved the problems I raised in the review. The manuscript can be considered for publication. Some prospects: 1. The effect of demonstrating the advantages of 35 nm extreme lithography resolution and device performance is still not good. I hope the author can find better devices to systemize this feature in the future. 2. The author proposed an additive manufacturing technology for printing semiconductor oxides. Some semiconductor and photonic devices with true 3D architecture should be selected to demonstrate the advantages of this method. Obviously, the manuscript lacks data.

Response:

We appreciate your comments and constructive prospects. As recommended, we will conduct some other 2D/3D devices in future, hope to achieve a better performance and further demonstrate the advantage of our work.

Reviewer #2 (Remarks to the Author):

In my opinion the authors have sufficiently addressed the reviewer comments. The manuscript can be accepted for publication. I have just some minor comments.

Response:

We appreciate your comments and recognition of our revision.

1) I suggest the authors to mention the film thickness earlier in the main text. This is a VERY thin film being used and most of the multiphoton lithography literature cited in the paper does not use anything close to that thin of a resist coating.

Response:

We appreciate your suggestions. As requested, we have provided the film thickness in the main text (Page 2, highlighted in blue).

2) It is not clear what Supplementary figure 15a is supposed to be. The authors call it an array of nuts. Do they mean bolts?

Response:

We appreciate your comments. We apologize that our previous statement may not have been clear. As you mentioned, they are indeed bolts (Supplementary Fig.15a), we have modified the description in the revised manuscript. Besides, we have provided more new 3D microstructures (that completed within the last two weeks) in Supplementary Fig.15, in order to make our work more convincing in terms of 3D printing capability.

Reviewer #3 (Remarks to the Author):

Most of the expressed remarks were addressed completely making the manuscript suitable for the publication in its current form.

Response:

We appreciate your comments and recognition of our revision.

One more note, Fig 2 (i) remains not updated with the current reference [32] benchmarking 60 nm achievement of year 2022. The inclusion of it was made in the Supplementary Table 2, but not here. This would make the inset picture complete indicating the consistent progress and most recent advancements in the rapidly evolving field. Thus I recommend updating it. Sorry, for not noticing it and pointing directly from the first round of review.

Response:

We appreciate your comments and kind remind. We have updated the Fig. 2i in the revised manuscript, in which the reference (benchmarking 60 nm achievement of year 2022) you noted has been added.